# CP-POL + PPI: Conformal Guarantees in Partially-Observed Label Space

**Christian Ngnie**                                          *ngnie.christian@yahoo.fr; christian@codeforafrica.org*
*Code For Africa, South-Africa*

**Reviewed on OpenReview:** *https://openreview.net/forum?id=GEy2BtBQKa*

## Abstract

We study Conformal Prediction (CP) in the practical and challenging regime where labeled training and calibration data observe only a subset of the label space. In this setting, classical Conformal guarantees no longer control marginal risk and naive unseen labels detection methods are either overconservative or uninformative. We introduce CP-POL, a simple operational pipeline that couples Split CP over observed labels with a calibrated novelty test and integrates Prediction-Powered Inference (PPI) for finite sample population estimation. We provide a non-asymptotic theory that (i) proves Le Cam impossibility result: novelty test from features alone is hopeless without structural assumptions, (ii) derives tight finite-sample coverage decompositions that isolate the role of the non-conforming event $s(X) > q$, (iii) gives Dvoretzky-Kiefer-Wolfowitz (DKW)-based conservative estimators and anytime martingale analogues for the novel mass function $\pi_{nov}$, (iv) identifies practically meaningful structural conditions under which strong guarantees for novel region prediction hold, and (v) proves finite-sample PPI bounds that cleanly separate sampling fluctuation, trained model error and novel-mass effects. We validate the theory with reproducible simulations. All bounds are non-asymptotic and designed for immediate use in deployed monitoring pipelines.

## 1 Introduction

Modern deployed classification systems increasingly face a simple-but-hard reality: the list of label categories that appears at training time is rarely the full label space the system will meet in the wild. Indeed, labels can be expensive to obtain, categories evolve and long-tailed or rare phenomena regularly create new labels. At the same time, practitioners demand calibrated and auditable behavior from models in such a way that when the system is unsure, it must either produce reliable set of candidate labels or clearly flag an input as "NOVEL" so that downstream decisions can proceed safely. This practical tension is the motivating problem of this paper.

Conformal Prediction (CP) is an attractive foundation for uncertainty quantification precisely because it supplies finite-sample and distributional-free marginal guarantees with minimal modeling assumptions. However, the canonical CP setup builds on the implicit assumption that the calibration data covers the label space, such that the labels the system will see at test time are represented in the calibration data. When that assumption fails, the standard CP framework can no longer provide marginal guarantees, and naive extensions either over-claim coverage or impose impractically large human-review burdens.

Related work spans several interconnected strands. CP has been extended in many directions (Vovk et al., 2005; Shafer & Vovk, 2008; Angelopoulos & Bates, 2023). Open-set and missing-mass formulations study uncertainty when new categories appear or when generative oracles are used to probe support (Noorani et al., 2025; Quach et al., 2023). Partial-label Conformal constructions treat per-example candidate sets that contain the true class (Javanmardi et al., 2023), a different data model from singleton annotated labels with partially-observed labels space. Recent work on calibrated semi-supervised and pseudo-label methods (Zhou et al., 2025), Rodriguez et al. (2023) investigates how unlabeled data and model pseudo-labels can be

safely combined with Conformal techniques. Parallel to this, the Prediction-Powered Inference (PPI) program (Angelopoulos et al., 2023b;a) develops finite-sample methods that exploit learned models to reduce variance in M-estimation while using a labeled anchor to protect against bias.

This work unifies these perspectives by anchoring Conformal calibration to human labels (preserving Split Conformal guarantees on the observed subpopulation) using model scores to define a principled novelty region, and applying PPI framework to produce provably conservative, finite-sample population estimates in partially-observed label space setting. First, we prove an important fact which is without structural assumptions linking features to novelty, detection of unobserved labels from features alone is fundamentally unidentifiable. Second, we develop a set of complementary positive results that are useful under realistic assumptions. We present finite-sample, non-parametric bounds allowing a practitioner to use a labeled calibration anchor together with an unlabeled feature pool to obtain conservative lower and upper bounds on quantities of interest. Third, we identify structural separation conditions that make guarantees possible. The first is margin separation condition: if novel-class examples uniformly acquire larger novelty scores than known-class examples above some margin, then a simple thresholding rule will reliably flag novel points and marginal CP coverage can be recovered. The second is stochastic dominance: if the tail of the novelty-score distribution under novel labels dominates the tail under known labels by a fixed factor, then we can translate an empirically-observed calibration false-positive rate into a quantitative lower bound on novelty detection power. Fourth, we develop a comprehensive PPI analysis tailored to the partially-observed label space setting. We show how to frame population estimation task under PPI one-step correction and provide non-asymptotic guarantees in both batch and sequential settings.

The results deliver a concrete toolkit for practitioners who deploy calibrated systems without a priori exhaustive label categories. Equally important, the paper clarifies the limits of what can be known and in the limited knowledge setting, provide as safe operational response to collect labels or enlarge the calibration anchor rather than over-trusting model predictions.

## 2 Setup and notation

Let $(X, Y) \sim P$ on $\mathcal{X} \times \mathcal{Y}$ with possibly large $\mathcal{Y}$. The observed labeled dataset $D = \{(X_i, Y_i)\}_{i=1}^n$ contains only a subset

$$\mathcal{Y}_{\text{obs}} := \{y \in \mathcal{Y} : \exists i \leq n \text{ s.t. } Y_i = y\}$$

Define the novel event $N_{ov} = \{Y \notin \mathcal{Y}_{\text{obs}}\}$ and novel mass $\pi_{\text{nov}} = P(N_{ov})$

We train a model $\widehat{f}(x) = (\widehat{f}_y(x))_{y \in \text{obs}}$ on a training split and hold out a calibration anchor $S_{\text{calib}}$ of size $m$, exchangeable with test points. For $y \in \mathcal{Y}_{\text{obs}}$ define the nonconformity scores:

$$a(x, y) := 1 - \widehat{f}_y(x) \in [0, 1], \qquad s(x) := \min_{y \in \mathcal{Y}_{\text{obs}}} a(x, y) = 1 - \max_y \widehat{f}_y(x).$$

Let $q$ be the $(1 - \alpha)$ empirical quantile of calibration nonconformities. The conformal set of known labels is

$$C_{\text{known}}(x; q) = \{y \in \mathcal{Y}_{\text{obs}} : a(x, y) \leq q\}.$$

Crucial identity:

$$C_{\text{known}}(x; q) = \varnothing \iff s(x) > q$$

We will study coverage, detection and estimation properties for $C_{\text{known}}$ and the novel region $S_t := \{x : s(x) > t\}$, and we will integrate PPI for population. parameters via M-estimators.

## 3 Theoretical framework

We present a unified framework that addresses both finite-sample uncertainty quantification and population parameter estimation in partially-observed label space.

### 3.1 Conformal Prediction with novelty detection

#### 3.1.1 Prediction sets and operational procedure

We define the overall prediction set $\widehat{C}(X)$ as follows:

**Definition 3.1** (CP-POL Prediction Set). *For a test sample $X$, the CP-POL prediction set is defined as:*

$$\widehat{C}(X) = \begin{cases} C_{\text{known}}(X;q) & \text{if } C_{\text{known}}(X;q) \neq \emptyset \\ \{\texttt{NOVEL}\} & \text{if } C_{\text{known}}(X;q) = \emptyset \text{ and } s(X) > t \\ \{\arg\max_{y \in \mathcal{Y}_{\text{obs}}} \widehat{f}_y(X)\} & \text{otherwise (fallback)} \end{cases}$$

This definition leads to the operational procedure:

---

**Algorithm 1** CP-POL: Conformal Prediction for Partially-Observed Labels

---

**Require:** Calibration set $S_{\text{calib}}$ of size $m$, imputer $\widehat{f}$, miscoverage level $\alpha \in (0,1)$, novelty threshold $t \in [0,1]$
**Ensure:** Prediction set $\widehat{C}(x)$ for new test point $x$

 1: Compute calibration nonconformity scores $a_i = 1 - \widehat{f}_{Y_i}(X_i)$ for $i \in S_{\text{calib}}$
 2: Set $q = $ empirical $(1-\alpha)$ quantile of $\{a_i\}_{i=1}^m$
 3: For new $x$: compute $C_{\text{known}}(x;q) = \{y \in \mathcal{Y}_{\text{obs}} : 1 - \widehat{f}_y(x) \leq q\}$
 4: **if** $C_{\text{known}}(x;q) \neq \varnothing$ **then**
 5:     **return** $\widehat{C}(x) = C_{\text{known}}(x;q)$             ▷ Prediction from known labels
 6: **else if** $s(x) > t$ **then**             ▷ Novelty score exceeds threshold
 7:     **return** $\widehat{C}(x) = \{\texttt{NOVEL}\}$              ▷ Flag as novel
 8: **else**
 9:     **return** $\widehat{C}(x) = \{\arg\max_{y \in \mathcal{Y}_{\text{obs}}} \widehat{f}_y(x)\}$     ▷ Fallback to top prediction
10: **end if**

---

#### 3.1.2 Coverage guarantees and fundamental limits

**Theorem 3.2** (Marginal guarantee for known labels (Vovk et al., 2005; Angelopoulos & Bates, 2023)). *If a test sample $(X,Y)$ is exchangeable with the calibration set $S_{\text{calib}}$ and $Y \in \mathcal{Y}_{\text{obs}}$, then:*

$$\Pr\left(Y \in C_{\text{known}}(x;q) \mid Y \in \mathcal{Y}_{\text{obs}}\right) \geq 1 - \alpha$$

The overall coverage decomposes into known and novel components:

**Theorem 3.3** (Global Coverage Decomposition). *For any novelty threshold $t$,*

$$\Pr\left(Y \in \widehat{C}(X)\right) \geq (1 - \pi_{\text{nov}})(1 - \alpha) + \pi_{\text{nov}} \text{TPR}(t),$$

*where $\text{TPR}(t) = \Pr(s(X) > t \mid N)$ is the true positive rate for novelty detection.*

*Proof.*

$$\begin{aligned} \Pr(Y \in \widehat{C}(X)) &= \Pr(Y \in \widehat{C}(X) \mid \neg N_{ov}) \cdot (1 - \pi_{\text{nov}}) + \Pr(Y \in \widehat{C}(X) \mid N_{ov}) \cdot \pi_{\text{nov}} \\ &= \Pr(Y \in C_{\text{known}}(X;q) \mid \neg N_{ov}) \cdot (1 - \pi_{\text{nov}}) + \Pr(\texttt{NOVEL} \in \widehat{C}(X) \mid N_{ov}) \cdot \pi_{\text{nov}} \\ &= \Pr(Y \in C_{\text{known}}(X;q) \mid \neg N_{ov}) \cdot (1 - \pi_{\text{nov}}) + \text{TPR}(t) \cdot \pi_{\text{nov}}. \end{aligned}$$

We get the result by applying Theorem 3.2. $\qquad\square$

Theorem 3.3 shows global coverage depends on $\text{TPR}(t)$. We state below the detection is impossible without linking test sample $X$ to label novelty. To understand that fundamental limit from features alone, we employ Le Cam's two-point method.

**Lemma 3.4** (Testing Lower Bound via Total Variation). *For any two distributions $P_0$ and $P_1$, and any test $\mathcal{A} : \Omega \to \{0, 1\}$, the sum of Type I and Type II errors is bounded below by:*

$$P_0(\mathcal{A} = 1) + P_1(\mathcal{A} = 0) \geq 1 - \delta_{TV}(P_0, P_1).$$

*In particular, if $\delta_{TV}(P_0, P_1) = 0$, then no test can outperform random guessing:*

$$\inf_{\mathcal{A}} \left[ P_0(\mathcal{A} = 1) + P_1(\mathcal{A} = 0) \right] \geq 1.$$

*Proof.* For any test $\mathcal{A}$, we have:

$$
\begin{aligned}
P_0(\mathcal{A} = 1) + P_1(\mathcal{A} = 0) &= 1 - \left[ P_0(\mathcal{A} = 0) - P_1(\mathcal{A} = 0) \right] \\
&\geq 1 - \sup_{A \in \mathcal{F}} |P_0(A) - P_1(A)| \\
&= 1 - \delta_{\mathrm{TV}}(P_0, P_1).
\end{aligned}
$$

$\square$

**Notation.** For any joint distribution $P$ on $\mathcal{X} \times \mathcal{Y}$ we denote by $P_X$ the marginal distribution of the feature $X$ (i.e. $P_X(A) = P(X \in A)$ for measurable $A \subseteq \mathcal{X}$). Similarly, for $j \in \{0, 1\}$ we write $P_j$ for a joint distribution on $\mathcal{X} \times \mathcal{Y}$ and $P_{j,X}$ for its $X$-marginal.

**Theorem 3.5** (Fundamental Impossibility). *For any measurable novelty detector $\mathcal{A} : \mathcal{X} \to \{\texttt{NOVEL}, \texttt{KNOWN}\}$ and any sample size $n$, there exist distributions $P_0, P_1$ on $\mathcal{X} \times \mathcal{Y}$ with:*

- *$P_{0,X} = P_{1,X}$ (identical feature marginals)*

- *Under $P_0$: $\pi_{\mathrm{nov}} = 0$, under $P_1$: $\pi_{\mathrm{nov}} > 0$*

*such that:*

$$\sup_{j \in \{0,1\}} \Pr_{P_j}(\mathcal{A}(X) \text{ is incorrect}) \geq \tfrac{1}{2}.$$

*Proof.* Let $P_X$ be any distribution on $\mathcal{X}$. We define $P_0$: $X \sim P_X$, with $Y \equiv y_0$ for some fixed $y_0 \in \mathcal{Y}_{\mathrm{obs}}$ and $P_1$: $X \sim P_X$ where $Y = y_0$ with probability $\pi_{\mathrm{nov}} = 0$ and $Y = y_{\mathrm{novel}}$ with probability $\pi_{\mathrm{nov}} > 0$. By construction, $P_{0,X} = P_{1,X} = P_X$. Therefore, $\delta_{\mathrm{TV}}(P_{0,X}, P_{1,X}) = 0$. By lemma 3.4, we deduce $P_0(\mathcal{A}(X) = \texttt{NOVEL}) + P_1(\mathcal{A}(X) = \texttt{KNOWN}) \geq 1$, which implies $\max\{P_0(\mathcal{A}(X) = \texttt{NOVEL}), P_1(\mathcal{A}(X) = \texttt{KNOWN})\} \geq \tfrac{1}{2}$. $\square$

### 3.1.3 Positive results under structural assumptions

While Theorem 3.5 shows that detection is impossible without assumptions, the following structural conditions, which are theoretical, enable detection. In practice, one can assess their plausibility using calibration diagnostics. For instance, comparing the empirical CDFs of novelty scores on calibration data (known) and an unlabeled pool (mixture) via two-sample tests can indicate separation. Moreover, the DKW lower bound for $\pi_{\mathrm{nov}}$ (Theorem 3.8) provides a conservative estimate. A positive lower bound suggests that the mixture distribution differs from the known distribution, hinting at detectable novelty.

**Theorem 3.6** (Perfect Detection Under Margin). *If there exists $\tau \in (0, 1]$ such that $s(X) \geq \tau$ almost surely for novel instances, and $t < \tau$, then $\mathrm{TPR}(t) = 1$ and CP-POL achieves coverage at least $1 - \alpha$.*

*Proof.* If for a novel $(X, Y)$ we have $s(X) \geq \tau$ and $t < \tau$, then every novel point satisfies $s(X) > t$, hence $\mathrm{TPR}(t) = 1$. The result follows from Theorem 3.3. $\square$

**Theorem 3.7** (Detection Under Stochastic Dominance). *If there exist $t_0$ and $\lambda > 1$ such that for all $u \geq t_0$:*

$$\frac{dF_{\mathrm{nov}}}{dF_{\mathrm{known}}}(u) \geq \lambda,$$

*then for $t \geq t_0$:*

$$\text{TPR}(t) \geq \frac{\lambda \cdot \text{FPR}(t)}{\lambda \cdot \text{FPR}(t) + 1 - \text{FPR}(t)},$$

*where* $\text{FPR}(t) = \Pr(s(X) > t \mid Y \in \mathcal{Y}_{\text{obs}})$.

*Proof.* For $u \geq t_0$, by definition of the likelihood ratio, $1 - F_{\text{nov}}(t) = \int_t^1 dF_{\text{nov}}(u) \geq \lambda \int_t^1 dF_{\text{known}}(u) = \lambda(1 - F_{\text{known}}(t))$. Rearranging using $\text{TPR}(t) = 1 - F_{\text{nov}}(t)$ and $\text{FPR}(t) = 1 - F_{\text{known}}(t)$ give the result. $\square$

### 3.2 Finite-sample estimation of $\pi_{\text{nov}}$ and novelty detection performance

The estimation of $\pi_{(nov)}$ serves crucial purposes in CP-POL pipeline:

- It quantifies the prevalence of novel labels at inference time, providing operators with a direct measure of how frequently the model encounters unobserved labels.

- It allows practitioners to understand and potentially adjust the actual coverage guarantees they can expect in practice ( 3.3).

- it helps determine the appropriate additional labeling budget.

#### 3.2.1 Theoretical lower bound of $\pi_{\text{nov}}$

Assume an unlabeled pool of size $N$, i.i.d. drawn from $P_X$. Let $\widehat{F}_{\text{known}}(u)$ and $\widehat{F}_{\text{mix}}(u)$, CDFs of nonconformity scores defined respectively on calibration and unlabeled pool. By the mixture identity,

$$F_{\text{mix}}(u) = (1 - \pi)F_{\text{known}}(u) + \pi F_{\text{nov}}(u) \leq (1 - \pi)F_{\text{known}}(u) + \pi,$$

so for any $u$,

$$\pi \geq \frac{F_{\text{mix}}(u) - F_{\text{known}}(u)}{1 - F_{\text{known}}(u)}. \tag{3.1}$$

#### 3.2.2 DKW-based finite-sample bounds

**Theorem 3.8** (Finite-sample DKW lower bound for $\pi_{\text{nov}}$). *Let $\delta \in (0, 1)$, $u$ such that $1 - F_{\text{known}}(u) > 0$ and $\epsilon := max(\sqrt{\frac{\log(4/\delta)}{2m}}, \sqrt{\frac{\log(4/\delta)}{2N}})$. With probability at least $1 - \delta$ over the joint draw of calibration and unlabeled pool,*

$$\pi \geq \frac{\widehat{F}_{\text{mix}}(u) - \widehat{F}_{\text{known}}(u) - 2\epsilon}{1 - \widehat{F}_{\text{known}}(u) - \epsilon},$$

*provided the denominator is positive.*

*Proof.* For an i.i.d. sample of size $N$ and any $\eta > 0$, DKW inequality implies

$$\Pr\left(\sup_x |\widehat{F}_N(x) - F(x)| > \eta\right) \leq 2e^{-2N\eta^2}.$$

By applying DKW separately to the calibration sample and unlabeled pool, with probability at least $1 - \delta$ we have simultaneously

$$|\widehat{F}_{\text{known}}(u) - F_{\text{known}}(u)| \leq \sqrt{\frac{\log(4/\delta)}{2m}} \leq \epsilon, \qquad |\widehat{F}_{\text{mix}}(u) - F_{\text{mix}}(u)| \leq \sqrt{\frac{\log(4/\delta)}{2N}} \leq \epsilon. \tag{3.2}$$

From equation 3.2 we have with probability at least $1 - \delta$ the conservative replacements

$$F_{\text{mix}}(u) \geq \widehat{F}_{\text{mix}}(u) - \epsilon, \qquad F_{\text{known}}(u) \leq \widehat{F}_{\text{known}}(u) + \epsilon.$$

Substituting these into equation 3.1 yields,

$$\pi \geq \frac{(\widehat{F}_{\text{mix}}(u) - \epsilon) - (\widehat{F}_{\text{known}}(u) + \epsilon)}{1 - (\widehat{F}_{\text{known}}(u) + \epsilon)} = \frac{\widehat{F}_{\text{mix}}(u) - \widehat{F}_{\text{known}}(u) - 2\epsilon}{1 - \widehat{F}_{\text{known}}(u) - \epsilon}.$$

$\square$

### 3.3 Sequential anytime bounds for streaming unlabeled pools

In many deployed settings, unlabeled features arrive sequentially and operators want at every time $t$ a valid lower bound $\underline{\pi}_t$ for the novel mass and corresponding guarantees for TPR at pre-defined threshold $t_0$. The DKW is not uniform in time, so we replace it with time-uniform confidence sequences for CDFs.

Let the unlabeled stream $X_1, X_2, \ldots$ from $P_X$, a threshold $u \in (0, 1]$, the scores $s_i := s(X_i)$ and the indicators $Z_i := \mathbb{1}\{s_i \leq u\}$. Let the Bernoulli mean $p = \mathbb{E}[Z_i]$ and the empirical mean $\widehat{p}_t = S_t/t$ where $S_t = \sum_{i=1}^t Z_i$. We want confidence sequences $[L_t(u), U_t(u)]$ such that:

$$\Pr\left(\forall t \geq 1: \ L_t(u) \leq F_{\mathrm{mix}}(u) \leq U_t(u)\right) \geq 1 - \delta.$$

#### 3.3.1 Confidence sequences for streaming CDF estimation

**Theorem 3.9** (Hoeffding-Style Confidence Sequence)**.** *Using error schedule $\delta_t = \frac{6\delta}{\pi^2 t^2}$ with $\sum_{t=1}^\infty \delta_t = \delta$, Hoeffding's inequality gives for each $t$,*

$$\Pr\left(\widehat{p}_t - p > \sqrt{\frac{\log(2/\delta_t)}{2t}}\right) \leq \delta_t,$$

*and a union bound over $t$ gives the time-uniform statement. Therefore, a valid one-sided lower CS is:*

$$L_t^{\mathrm{mix}}(\delta) = \max\left\{0, \widehat{p}_t - \sqrt{\frac{\log(2/\delta_t)}{2t}}\right\}.$$

**Theorem 3.10** (KL-Inversion Confidence Sequence ((Howard et al., 2021)))**.** *A tighter one-sided lower CS $L_t^{\mathrm{KL}}(\delta)$ is the largest $q \in [0, \widehat{p}_t]$ such that:*

$$t \cdot \mathrm{KL}(\widehat{p}_t \| q) \leq \log\left(\frac{c(t+1)}{\delta}\right),$$

*where $\mathrm{KL}(x\|y)$ is the binary KL divergence and $c \geq 1$ is a constant.*

**Theorem 3.11** (Variance-Adaptive Empirical Bernstein Confidence Sequence)**.** *Let $\widehat{V}_t = \frac{1}{t}\sum_{i=1}^t (Z_i - \widehat{p}_t)^2$ the sample variance. A valid one-sided lower CS is:*

$$L_t^{\mathrm{EB}}(\delta) = \widehat{p}_t - \sqrt{\frac{2\widehat{V}_t \log(3/\delta_t)}{t}} - \frac{7\log(3/\delta_t)}{3(t-1)},$$

*where $\delta_t = \frac{6\delta}{\pi^2 t^2}$ for $t \geq 2$.*

**Comparison:**

- **Hoeffding (Theorem 3.9)**: Most conservative, simplest to implement, ideal for initial prototyping

- **KL-inversion (Theorem 3.10)**: Tighter bounds for probabilities near 0 or 1, requires numerical root-finding

- **Empirical Bernstein (Theorem 3.11)**: Variance-adaptive, particularly effective for right-tail detection where variance is small, ideal for production use when computational overhead is acceptable

#### 3.3.2 Time-Uniform Novelty Detection

Combining streaming CDF bounds with fixed calibration bounds:

**Theorem 3.12** (Time-Uniform Lower Bound for $\pi_{\mathrm{nov}}$)**.** *With probability at least $1 - \delta$, for all $t \geq 1$ simultaneously:*

$$\pi_{\mathrm{nov}} \geq \max\left\{0, \frac{L_t(u) - U_{\mathrm{known}}(u)}{1 - U_{\mathrm{known}}(u)}\right\},$$

*where $U_{\mathrm{known}}(u) = \min\left\{1, \widehat{F}_{\mathrm{known}}(u) + \sqrt{\frac{\log(2/\delta_{\mathrm{cal}})}{2m}}\right\}$ and $L_t(u)$ can be any of the confidence sequences from Theorems 3.9, 3.10, or 3.11.*

# 4 Prediction-Powered Inference (PPI) for partially-observed label space

Prediction-Powered Inference introduced by Angelopoulos et al. (2023b) leverages predictions from a trained model to improve statistical efficiency while preserving valid finite-sample inference. In the standard setting, one observes: (i) a large unlabeled sample $\{X_i\}_{i=1}^N$ from a target population, (ii) a smaller labeled subsample $\{X_i, Y_i\}_{i \in \mathcal{L}}$ of size $n \ll N$, and (iii) a predictive model producing $\widehat{Y}_i$ for all $i$. PPI constructs an estimator by combining a large-sample plug-in term based on model predictions and a labeled-sample correction term that removes bias. For convex risk minimization problems, the authors define a measure-of-fit term $m_\theta$ computed on the large unlabeled set using model predictions, and a rectifier $\Delta_\theta$ computed on the labeled sample. Their inference procedure is base on the quantity

$$m_\theta + \Delta_\theta.$$

Our goal is to extend this framework to the partially-observed label space, providing finite-sample guarantees for general M-estimators in both batch and sequential settings. The central idea is to leverage a trained model to approximate conditional expectations in estimating equations, calibrate via the labeled sample to remove bias and explicitly account for novel-label through $\pi_{(nov)}$.

## 4.1 M-estimation formulation

Let $Z = (X, Y)$ and suppose the target parameter $\theta^\star$ satisfies

$$\mathbb{E}[\Psi(Z; \theta^\star)] = 0.$$

If all labels were observed, the empirical classical M-estimator would solve

$$\frac{1}{N} \sum_{i=1}^N \Psi(Z_i; \widehat{\theta}) = 0.$$

In our partially-observed setting, we observe $X_1, \ldots, X_N$, and for each $i$ an indicator $\xi_i \in \{0, 1\}$ denotes whether $Y_i$ is observed. We assume the sampling probability $\pi_i = \mathbb{P}(\xi_i = 1)$ is known.

Define

$$m(x; \theta) := \mathbb{E}[\Psi_{\mathrm{obs}}(Y, X; \theta) \mid X = x].$$

Following the PPI principle, we approximate $m(x; \theta)$ with a predictive model $\widehat{m}(x; \theta)$ trained on the labeled sample.

We define the PPI contribution:

$$\widehat{\Psi}_i(\theta) = \widehat{m}(X_i; \theta) + \frac{\xi_i}{\pi_i} \big( \Psi_{\mathrm{obs}}(Y_i, X_i; \theta) - \widehat{m}(X_i; \theta) \big). \tag{4.1}$$

The estimator $\widehat{\theta}$ solves

$$\frac{1}{N} \sum_{i=1}^N \widehat{\Psi}_i(\widehat{\theta}) = 0. \tag{4.2}$$

## 4.2 Connection to Standard PPI Formulation

Equation equation 4.2 is algebraically equivalent to the standard PPI contribution. Indeed, writing $\mathcal{L}$ for the labeled set (size $n$), and using $\pi_i = n/N$ under uniform sampling, one obtains:

$$\frac{1}{N} \sum_{i=1}^N \widehat{\Psi}_i(\theta) = \underbrace{\frac{1}{N} \sum_{i=1}^N \widehat{m}(X_i; \theta)}_{m_\theta} + \underbrace{\frac{1}{n} \sum_{i \in \mathcal{L}} \big( \Psi_{\mathrm{obs}}(Y_i, X_i; \theta) - \widehat{m}(X_i; \theta) \big)}_{\Delta_\theta}.$$

Thus, the empirical estimating equation coincides with the PPI quantity $m_\theta + \Delta_\theta$ used in Angelopoulos et al. (Angelopoulos et al. (2023b)). In the special case of mean estimation, this reduces exactly to their closed-form PPI estimator:

$$\widehat{\theta} = \frac{1}{N}\sum_{i=1}^{N}\widehat{Y}_i + \frac{1}{n}\sum_{i\in\mathcal{L}}(Y_i - \widehat{Y}_i).$$

Our formulation therefore generalizes standard PPI to arbitrary M-estimation problems under partial label observation.

### 4.3 Finite-sample guarantees in batch setting

**Assumption 4.1** (Boundedness and approximation). *There exists a constant $B > 0$ such that for all $\theta$ in a neighbourhood of $\theta^\star$ and all $(x, y)$*

$$\|\Psi_{\mathrm{obs}}(y, x; \theta)\| \le B, \qquad \|\widehat{m}(x; \theta) - m(x; \theta)\| \le \Delta(x),$$

*where $\Delta : \mathcal{X} \to [0, B]$ is measurable.*

**Assumption 4.2** (Jacobian invertibility). *Let $J(\theta) := \mathbb{E}[\nabla_\theta \Psi(Z; \theta)]$ denote the population Jacobian. There exist constants $\kappa < \infty$ and $L < \infty$ such that*

$$\|J(\theta^\star)^{-1}\|_{\mathrm{op}} \le \kappa, \qquad \Psi(\cdot, \cdot; \theta) \text{ is } L\text{-Lipschitz in } \theta \text{ near } \theta^\star.$$

**Assumption 4.3** (Sampling / IPW / cross-fitting). *The data $\{(X_i, N_i, Y_i, \xi_i)\}_{i=1}^{N}$ are i.i.d. The sampling mechanism satisfies:*

*(i) For each $i$, conditional on $X_i$ the labeling indicator $\xi_i$ is independent of $Y_i$: $\xi_i \perp Y_i \mid X_i$.*

*(ii) $\mathbb{E}[\xi_i/\pi_i \mid X_i] = 1$ a.s.*

*(iii) There exists $\pi_{\min} > 0$ with $\pi_i \ge \pi_{\min}$ a.s., hence $\xi_i/\pi_i \le 1/\pi_{\min}$ a.s.*

*The nuisance $\widehat{m}(\cdot; \theta^\star)$ is constructed by $K$-fold cross-fitting and satisfies the uniform mean-bias control*

$$\sup_x \left\| \mathbb{E}[\widehat{m}^{(-k)}(x; \theta^\star)] - m(x) \right\| \le \eta_N,$$

*with deterministic $\eta_N \ge 0$ and $\eta_N \le \bar{\Delta}_N$ wher $\bar{\Delta}_N := \frac{1}{N}\sum_{i=1}^{N}\|\widehat{m}(X_i; \theta^\star) - m(X_i; \theta^\star)\|$. .*

**Theorem 4.4** (PPI finite-sample bound — batch). *Under Assumptions 4.1, 4.2 and 4.3, let $\widehat{\theta}$ solve*

$$\frac{1}{N}\sum_{i=1}^{N}\widehat{\Psi}_i(\widehat{\theta}) = 0, \qquad \widehat{\Psi}_i(\theta) := \widehat{m}(X_i; \theta) + \frac{\xi_i}{\pi_i}\big(\Psi_{\mathrm{obs}}(Y_i, X_i; \theta) - \widehat{m}(X_i; \theta)\big).$$

*Then for any $\delta \in (0, 1)$, with probability at least $1 - \delta$,*

$$\|\widehat{\theta} - \theta^\star\| \le 2\kappa \Big( C B \sqrt{\frac{\log(2p/\delta)}{N}} + \bar{\Delta}_N + B\,\pi_{\mathrm{nov}} \Big), \tag{4.3}$$

*where $C = 3\,(1 + 1/\pi_{\min})$.*

**Discussion** The finite-sample bound in Theorem 4.4 rests on three stated structural assumptions. First, the PW conditions ($\xi \perp Y \mid X$, known inclusion probabilities $\pi_i$, and a positive lower bound $\pi_{\min}$) are used only to guarantee unbiasedness of the IPW correction and to control the magnitude of the weights in concentration inequalities. In practice extremely small $\pi_i$ blow up the stochastic term, which is reflected quantitatively by the $(1 + 1/\pi_{\min})$ factor in the constant $C$. Second, the Jacobian invertibility and local Lipschitz conditions provide the classical M-estimator regularity needed to convert a uniform concentration bound on the empirical

PPI moment into a bound on the parameter error via a one-step mean-value expansion. The inverse Jacobian norm $\kappa$ therefore scales every error contribution and encodes local stability of the estimating equation. Third, the boundedness packages the cost of replacing conditional expectations with predictions: the averaged approximation error $\bar{\Delta}_N$ is the explicit price paid for using a plug-in model and is the only term in the bound that does not vanish automatically with $N$ unless the model is consistent fast enough. These three ingredients map directly onto the three terms in equation 4.3: (i) a concentration term of order $\mathcal{O}(N^{-1/2})$ inflated by IPW weights, (ii) the model approximation penalty $\bar{\Delta}_N$, and (iii) a novelty penalty proportional to $\pi_{\mathrm{nov}}$ that captures irreducible bias when parts of the label space were never observed during training. Unlike the classical PPI exposition which emphasizes a convex-risk, grid-testing recipe built from $m_\theta + \Delta_\theta$, our statement treats a general M-estimation moment and hence replaces convexity assumptions by the standard Jacobian/Lipschitz regularity while making the approximation error explicit and finite-sample. Operationally, this gives clear guidance: (a) use cross-fitting or honest sample splitting so that the nuisance term behaves like a higher-order remainder (we impose a uniform mean-bias control to separate deterministic bias from stochastic fluctuation), (b) avoid designs with very small $\pi_i$ (or stabilize IPW weights) to limit inflation of the sampling term, and (c) monitor $\bar{\Delta}_N$ and $\pi_{\mathrm{nov}}$: when $\bar{\Delta}_N = o(N^{-1/2})$ and $\pi_{\mathrm{nov}} = o(N^{-1/2})$ the bound collapses to the usual root-$N$ asymptotic regime and the oracle PPI variance is recovered. Conversely, if the model error dominates the sampling term the estimator's performance is governed by $\bar{\Delta}_N$ and the user should prioritize improving the nuisance model or expanding labeled coverage. Finally, the explicit constants in the bound (notably $C = 3(1 + 1/\pi_{\mathrm{min}})$ and the Jacobian factor $\kappa$) are conservative artifacts of the concentration proofs. They reveal qualitatively how design (through $\pi_{\mathrm{min}}$) and local curvature (through $\kappa$) affect finite-sample behaviour and therefore provide immediate, actionable diagnostics for practitioners.

### 4.4 Sequential PPI with anytime guarantees

In streaming settings, we extend PPI to provide time-uniform guarantees using martingale concentration. We assume the data arrive sequentially. At each time $t$, we observe $X_t$, then $\xi_t$ and possibly $Y_t$ if $\xi_t = 1$. We maintain a sequence of estimators $\widehat{\theta}_T$ solving $\frac{1}{T}\sum_{t=1}^{T}\widehat{\Psi}_t(\widehat{\theta}_T) = 0$. Under assumptions analogous to the batch case, we obtain both fixed-time and time-uniform bounds.

Let $(\Omega, \mathcal{F}, \mathbb{P})$ be a probability space, $\{\mathcal{F}_t\}_{t\geq 0}$ a filtration with $\mathcal{F}_0 = \{\varnothing, \Omega\}$. For each $t \geq 1$, let the random objects

$$X_t,\ N_t \in \{0,1\},\ Y_t,\ \xi_t \in \{0,1\},$$

on $(\Omega, \mathcal{F}, \mathbb{P})$. At time $t$:

(S1) Given $\mathcal{F}_{t-1}$, the covariate $X_t$ is *realized* (not assumed $\mathcal{F}_{t-1}$–measurable).

(S2) Conditional on $(\mathcal{F}_{t-1}, X_t)$ the triplet $(N_t, Y_t, \xi_t)$ is drawn according to a conditional distribution that may depend on $(\mathcal{F}_{t-1}, X_t)$.

(S3) The observed record at time $t$ is $O_t := (X_t,\ \xi_t,\ Y_t\mathbf{1}\{\xi_t = 1\})$. The filtration is updated by

$$\mathcal{F}_t := \sigma\big(\mathcal{F}_{t-1} \cup \sigma(O_t)\big).$$

**Assumption 4.5** (Weight unbiasedness). *For every $t \geq 1$ the sampling indicator $\xi_t$ and the inclusion probability $\pi_t$ satisfy*

$$\mathbb{E}\Big[\frac{\xi_t}{\pi_t} \ \Big|\ \mathcal{F}_{t-1}, X_t\Big] = 1, \qquad \xi_t \perp (Y_t, N_t) \ \big|\ (\mathcal{F}_{t-1}, X_t).$$

*Equivalently $P(\xi_t = 1 \mid \mathcal{F}_{t-1}, X_t, Y_t, N_t) = P(\xi_t = 1 \mid \mathcal{F}_{t-1}, X_t) =: \pi_t$.*

**Assumption 4.6** (Nuisance conditional-mean control). *The nuisance estimator's conditional bias is controlled on average:*

$$\frac{1}{T}\sum_{t=1}^{T} \Big\|\mathbb{E}\big[\widehat{m}(X_t; \theta^\star) - m(X_t; \theta^\star) \mid \mathcal{F}_{t-1}\big]\Big\| \leq \overline{\Delta}_T.$$

**Assumption 4.7** (Novelty frequency). *Let the novelty probability at time $t$*

$$\pi_{\mathrm{nov}, t} := \Pr(N_t = 1 \mid \mathcal{F}_{t-1}).$$

*There exists $\pi_{\text{nov}} \in (0, 1)$ with $\pi_{\text{nov},t} \leq \pi_{\text{nov}}$ for all $t$.*

**Theorem 4.8** (PPI Sequential Estimation)**.** *Assume 4.1, 4.2, 4.5, 4.6 and 4.7. Let the martingale differences $\Delta_t(\theta) := \widehat{\Psi}_t(\theta) - \mathbb{E}[\widehat{\Psi}_t(\theta) \mid \mathcal{F}_{t-1}]$ satisfy $\|\Delta_t(\theta)\| \leq B$ a.s. Define*

$$\overline{\Delta}_T := \frac{1}{T} \sum_{t=1}^{T} \|\widehat{m}(X_t; \theta^\star) - m(X_t; \theta^\star)\|,$$

*and*

$$V_T(\theta) := \max_{1 \leq j \leq p} \sum_{t=1}^{T} \mathbb{E}\left[\Delta_{t,j}(\theta)^2 \mid \mathcal{F}_{t-1}\right] = \sum_{t=1}^{T} \mathbb{E}[\|\Delta_t(\theta)\|^2 \mid \mathcal{F}_{t-1}].$$

***Fixed-time guarantee:*** *For any $T \geq 1$, $\delta \in (0, 1)$, with probability at least $1 - \delta$,*

$$\|\widehat{\theta}_T - \theta^\star\| \leq 2\kappa\left(\sqrt{\frac{2V_T(\theta^\star)\log(2p/\delta)}{T^2}} + \frac{B\log(2p/\delta)}{3T} + \overline{\Delta}_T + B\,\pi_{\text{nov}}\right).$$

***Time-uniform guarantee:*** *For any $\delta \in (0, 1)$, with probability at least $1 - \delta$, uniformly for all $T \geq 1$,*

$$\|\widehat{\theta}_T - \theta^\star\| \leq 2\kappa\left(\sqrt{\frac{2V_T(\theta^\star)\log(2p/\delta_T)}{T^2}} + \frac{B\log(2p/\delta_T)}{3T} + \overline{\Delta}_T + B\,\pi_{\text{nov}}\right),$$

*where $\delta_T = 6\delta/(\pi^2 T^2)$.*

**Discussion** The sequential theorem trades static concentration for predictable-variance martingale control and mirrors the same three-term structure while providing time-uniform (anytime) guarantees useful for sequential monitoring and stopping rules. Assumptionally, the streaming setup replaces i.i.d. sampling with conditional unbiasedness of the IPW weights relative to the filtration (so $\mathbb{E}[\xi_t/\pi_t \mid \mathcal{F}_{t-1}, X_t] = 1$) and requires uniform boundedness of the single-step martingale increments to apply Bernstein/Hoeffding–type martingale inequalities. These conditions ensure the predictable variance process $V_T(\theta^\star)$ accurately summarizes the accumulated stochastic variability and permits self-normalized control of the empirical PPI moment. The online nuisance condition is the time-conditional analogue of cross-fitting: instead of independence we require that the average conditional bias of the online or cross-fitted nuisance be small (quantified by $\overline{\Delta}_T$), which ensures the nuisance term remains a lower-order correction when $\overline{\Delta}_T = o(T^{-1/2})$. Interpretationally, the fixed-time bound decomposes into (i) a martingale variance term scaling like $T^{-1/2}$ (with predictable variance $V_T/T^2$), (ii) a Bernstein higher-order term of order $T^{-1}$, (iii) the averaged nuisance bias, and (iv) the novelty penalty $B\,\pi_{\text{nov}}$. The time-uniform variant is obtained by the usual $\delta_T$ scheduling (we use $\delta_T = 6\delta/(\pi^2 T^2)$) and therefore yields valid uniform control over all stopping times—this is useful in practice for designing sequential tests or anytime confidence intervals that incorporate PPI augmentation. Practically, the sequential result implies the same operational guidelines as in batch: stabilize IPW weights to control variance inflation, enforce online debiasing / cross-fitting to keep nuisance bias negligible, and monitor novelty frequency since a persistent high $\pi_{\text{nov}}$ produces a constant bias floor regardless of horizon. Limitations worth noting: if inclusion probabilities or the labeling mechanism can adversarially depend on history in ways that violate weight-unbiasedness, our martingale control no longer holds and a modeling of the sampling mechanism is necessary. Similarly, the boundedness assumption on increments may be relaxed to sub-Gaussian tails at the expense of more elaborate predictable-variance control. Overall, the sequential theorem extends PPI-style variance reduction to streaming problems while preserving explicit finite-sample diagnostics (through $V_T$, $\overline{\Delta}_T$, $\pi_{\text{nov}}$, and $\pi_{\text{min}}$) that practitioners can compute and monitor in real time to understand when PPI provides reliable efficiency gains versus when the approximation or novelty terms dominate.

## 5 Empirical Validation

We conduct comprehensive experiments to validate the theoretical results (Theorems 3.5, 3.6, 3.7, 4.4, 4.8) from our frameworks. They systematically examine detection power, finite-sample coverage, bias-variance tradeoffs and sequential derivations under conditions that mirror real-world partially observed label settings.

### 5.1 Novelty Detection and DKW Bounds

#### 5.1.1 Experimental Design

We simulate scalar novelty scores $s(X)$, with parameters spanning regimes from Theorem 3.5's impossibility to Theorems 3.6-3.7's detectable regimes. Calibration samples ($m = 5000$) are drawn from known distribution $F_{\text{known}} \sim \text{Beta}(2,5)$ for realistic right-tail behavior, while unlabeled pools ($N = 10000$) follow mixture $(1-\pi)F_{\text{known}} + \pi F_{\text{nov}}$ with $\pi \in [0, 0.15]$. We examine three separation regimes, each corresponding to a theoretical condition:

- No separation ($F_{\text{nov}} = F_{\text{known}}$): Tests Theorem 3.5's impossibility

- Moderate separation ($F_{\text{nov}} = \text{Beta}(4,3)$): Tests Theorem 3.7's detectable regime

- Strong separation ($F_{\text{nov}} = \text{Beta}(6,2)$): Validates Theorem 3.6's perfect detection

#### 5.1.2 Results and Analysis

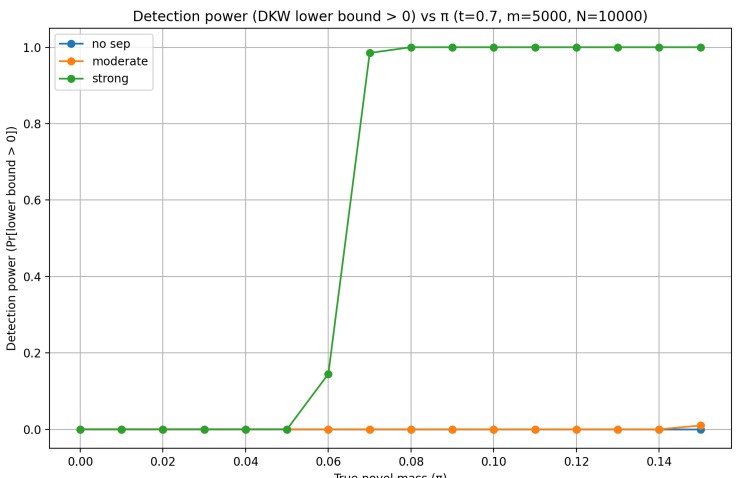

(a) Detection power vs true $\pi_{\text{nov}}$ across separation regimes. Dashed line shows theoretical detection boundary.

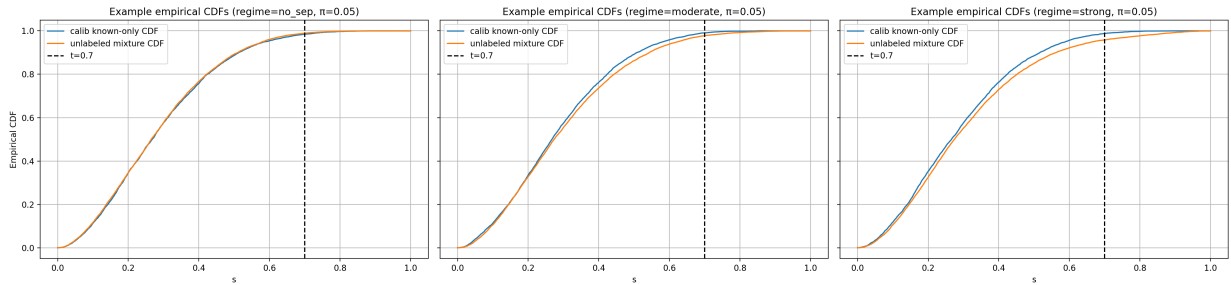

(b) Empirical CDFs illustrating separation regimes at $\pi = 0.05$, showing detectability conditions.

Figure 1: CP-POL detection performance validating theoretical predictions. (a) Detection power increases with separation strength and novel mass, confirming Theorems 3.6-3.7. Under no separation, detection power remains near zero, consistent with Theorem 3.5 (b) CDF separation enables detection under moderate and strong regimes.

**Fundamental Limits**: Under no separation, detection power remains near zero even for large $\pi_{\text{nov}} = 0.15$ and $N = 10000$, empirically validating Theorem 3.5's impossibility result. This confirms that without structural assumptions linking features to novelty, detection from features alone is fundamentally limited.

**Separation Enables Detection**: Under moderate and strong separation, detection power increases substantially with both $\pi_{\text{nov}}$ and separation strength (Figure 1a). The strong separation regime achieves near-perfect detection ($> 95\%$ power) for $\pi_{\text{nov}} \geq 0.05$, demonstrating the practical relevance of our margin condition. Figure 1b illustrates the CDF separation underlying this detectability.

**DKW Bound Performance**: Our DKW-based lower bounds maintain validity across all conditions, with coverage exceeding 95%. The bounds show expected conservatism that decreases with larger sample sizes, making them increasingly informative in data-rich regimes while maintaining theoretical guarantees.

**Threshold Selection and Effect on Calibration**: In practice, the novelty threshold $t$ can be selected directly from the calibration scores to target a user-specified false positive rate (FPR) on known data. Concretely, for a desired FPR $= q$ (e.g., $q \in \{0.01, 0.05, 0.10\}$), set

$$t \; := \; \widehat{F}_{\text{known}}^{-1}(1 - q),$$

where $\widehat{F}_{\text{known}}$ is the empirical CDF of calibration scores. This choice directly impacts the overall coverage via Theorem 3.3: decreasing $q$ (i.e., raising $t$) reduces the rate of NOVEL flags on known data (lower FPR) but may decrease TPR, thereby affecting the second term in the coverage bound. In our experiments, we find that $q = 0.1$ provides a good balance, achieving TPR around 0.5 in moderate separation and near 1.0 in strong separation, while maintaining high coverage on know classes.

### 5.1.3 Real-Data Validation on CIFAR-100 with Withheld Classes

We complement the synthetic experiments above with experiment on CIFAR-100, where we treat a subset of classes as *observed* and the remaining classes as *novel*. We train a ResNet-18 classifier on the observed classes and compute novelty scores on a held-out calibration split of observed-class examples.

**Novelty scores (MSP and Energy)**: We evaluate two widely-used uncertainty scores for novelty detection: (i) *maximum softmax probability* (MSP), using novelty score $-\max_k \widehat{p}_k(x)$, and (ii) the *energy score*, using novelty score $E(x)$ with $E(x) = -T \log \sum_k \exp(f_k(x)/T)$.

**Operational thresholding**: Following the heuristic described in Sec. 5.1, we choose $t$ by targeting a calibration-set FPR level $q \in \{0.01, 0.05, 0.10\}$ and report the achieved TPR for detecting withheld classes.

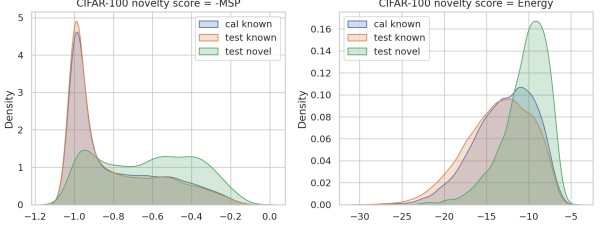 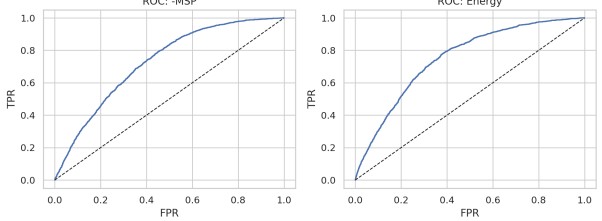

(a) Score distributions on CIFAR-100 for known vs withheld classes (MSP and Energy).

(b) ROC curves for novelty detection using MSP and Energy.

Figure 2: CIFAR-100 withheld-class experiment. Both MSP and Energy exhibit partial separation between known and withheld classes, illustrating the practical regime where novelty detection is feasible but imperfect; calibration-based thresholding enables an operational trade-off via target FPR.

As shown in Figure 2, both MSP and Energy scores exhibit partial separation between known and novel classes. Using the target FPR $q = 0.05$, we achieve TPR of approximately 0.14 (MSP) and 0.15 (Energy). This non-zero detection rate confirms that CP-POL can provide actionable novelty signals even in challenging real-world datasets, while maintaining valid coverage on known classes (as verified by coverage checks in Section 5.2).

### 5.2 Comparative Analysis: CP-POL vs. Classical Conformal Methods

We conduct comprehensive benchmark experiments comparing CP-POL against **Split CP (APS)** (Romano et al. (2020)), **RAPS** (Angelopoulos et al. (2021)), **TPS/Top-k** (Sadinle et al. (2019)), **Jackknife+** (Barber et al. (2021)) on both synthetic and real-world datasets. These methods are representative of state-of-the-art conformal prediction. None of them explicitly handle novel classes, serving as baselines to highlight the advantages of CP-POL.

#### 5.2.1 Synthetic Experiments

We extend our synthetic novelty-score experiments to evaluate all methods under the three separation regimes (no separation, moderate, strong) with 150 trials per configuration. The experimental setup uses 20 observed classes, 5 novel classes, 500 calibration samples, and test splits of 500 observed and 100 novel examples.

Table 1: Comparative Performance on Synthetic Data Across Separation Regimes ($\alpha = 0.1$)

(a) Coverage on Known Classes

| Method | No Separation | Moderate | Strong |
|---|---|---|---|
| CP-POL (Ours) | 0.948 | 0.948 | 0.948 |
| Jackknife+ | 1.000 | 1.000 | 1.000 |
| RAPS | 1.000 | 1.000 | 1.000 |
| Split CP (APS) | 1.000 | 1.000 | 1.000 |
| TPS/Top-k | 1.000 | 1.000 | 1.000 |
| Target (1-$\alpha$) | | 0.900 | |

(b) Novel detection rate

| Method | No Separation | Moderate | Strong |
|---|---|---|---|
| CP-POL (Ours) | **0.052** | **1.000** | **1.000** |
| Jackknife+ | 0.000 | 0.000 | 0.000 |
| RAPS | 0.000 | 0.000 | 0.000 |
| Split CP (APS) | 0.000 | 0.000 | 0.000 |
| TPS/Top-k | 0.000 | 0.000 | 0.000 |
| Target (1-$\alpha$) | | 0.900 | |

(c) Average Prediction Set Size (Known Classes)

| Method | No Separation | Moderate | Strong |
|---|---|---|---|
| CP-POL (Ours) | **1.00** | **1.00** | **1.00** |
| Jackknife+ | 1.00 | 1.00 | 1.00 |
| RAPS | 1.00 | 1.00 | 1.00 |
| Split CP (APS) | 3.099 | 3.099 | 3.099 |
| TPS/Top-k | 1.197 | 1.197 | 1.197 |

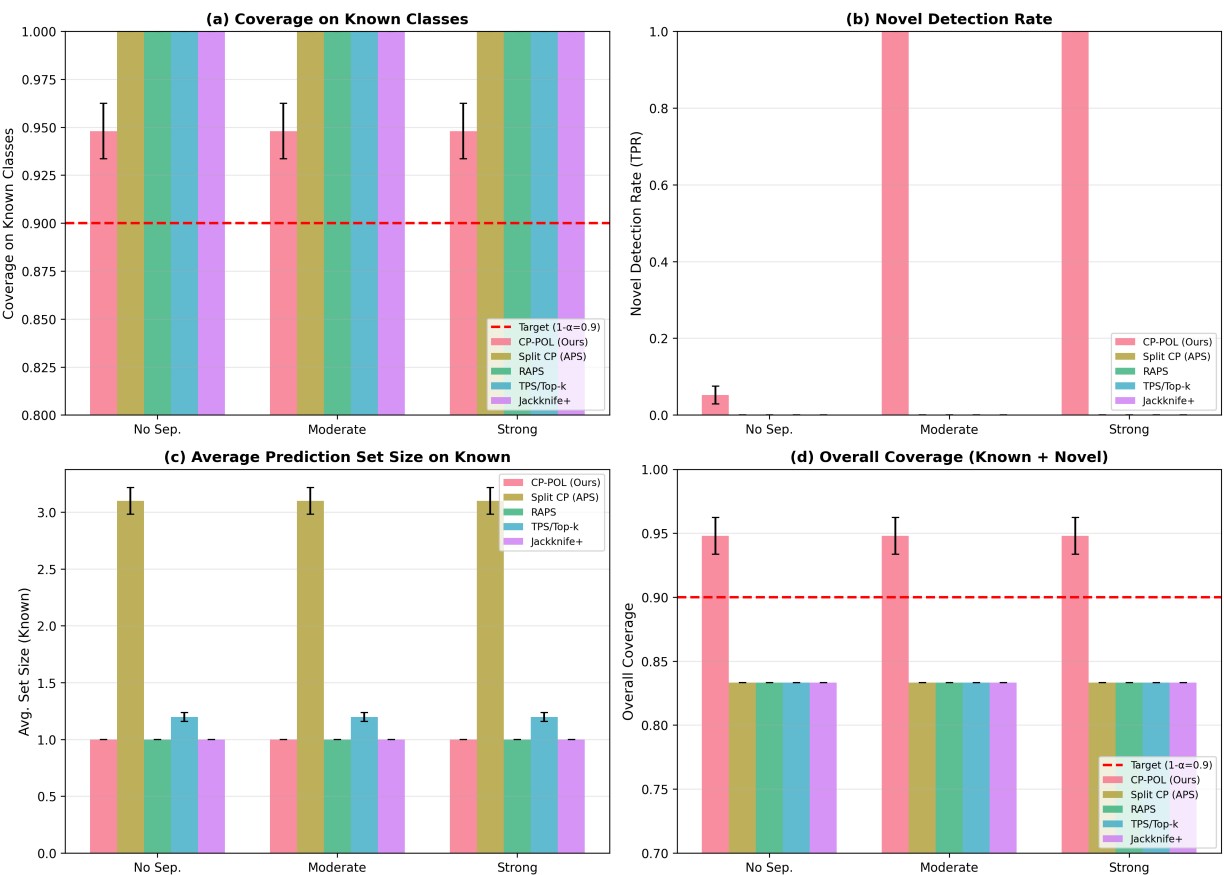

Figure 3: Comprehensive comparison of CP-POL vs. classical conformal methods on synthetic data. (a) Coverage on known classes: All the methods maintain target coverage across all regimes. (b) Novel detection rate: Our method outperforms the other methods in covering novel classes in moderate and strong separation regimes, but fails as the other methods with no separation regime, confirming Theorem 3.5. (c) Average set sizes: All the methods produce relatively small sets. (d) Overall coverage accounting for novel detection.

**Key Observations:**

1. **Coverage Validity**: All methods, including CP-POL and the baseline approaches, maintain the target coverage rate ($1-\alpha=0.9$) on known classes.

2. **Novel Detection Capability**: CP-POL demonstrates significant novel detection rates that vary with separation strength: perfect detection (100%) in moderate and strong separation regimes, and 5.2% detection in the no-separation regime (consistent with Theorem 3.5: detection is impossible without separation). In stark contrast, all baseline methods show 0% novel detection across all regimes, highlighting their fundamental limitation in open-world scenarios.

3. **Set Size Efficiency**: CP-POL, Jackknife+, RAPS and TPS/Top-k achieve efficient singleton or near-singleton predictions (avg. size =1-1.2) for known classes while maintaining coverage validity. Split CP (APS) produces larger prediction sets (avg. size =3.1) due to its different calibration approach.

### 5.2.2 CIFAR-100 Withheld-Classes Experiments

We extend the CIFAR-100 experiment to compare all methods using both MSP and Energy novelty scores. The setup withholds 20 classes as novel, trains on 80 observed classes, and evaluates on separate test sets.

Table 2: Comparative Performance on CIFAR-100 Withheld-Classes ($\alpha = 0.1$, Target FPR=0.05)

| Method | Known Coverage | | Known Avg. Set Size | | Novel Detection | |
|---|---|---|---|---|---|---|
| | MSP | Energy | MSP | Energy | MSP | Energy |
| CP-POL (Ours) | 0.856 | 0.861 | 2.14 | 2.19 | **0.136** | **0.151** |
| Jackknife+ | 0.709 | 0.709 | 1.00 | 1.00 | 0.000 | 0.000 |
| TPS/Top-k | 0.89 | 0.89 | 2.05 | 2.05 | 0.000 | 0.000 |
| RAPS | 0.709 | 0.709 | 1.00 | 1.00 | 0.000 | 0.000 |
| Split CP (APS) | 0.99 | 0.99 | 20.19 | 20.19 | 0.000 | 0.000 |
| Target (1-$\alpha$) | 0.900 | | – | | – | |

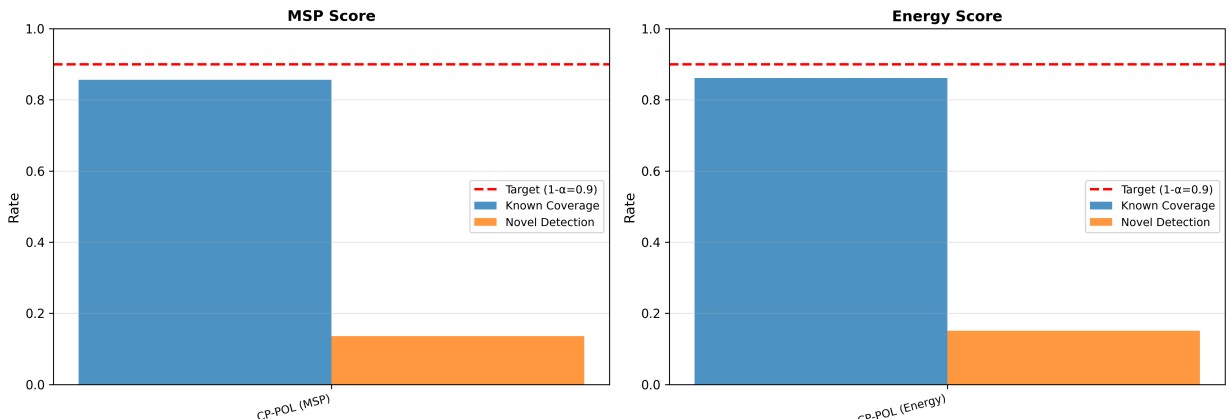

Figure 4: CIFAR-100 comparative results. CP-POL with Both MSP and Energy scores maintains high known covergae (>85%) and achieves measurable novel detection rate ( 14-15%), unlike baseline methods which fail completely at novel detection. Among baselines, known coverage varies but all show zero novel detection.

**Key Observations:**

1. **Coverage Breakdown in Classical Methods**: All baseline methods exhibit a complete failure in novel detection (0%). While known coverage varies, these methods are not designed to adapt to out-of-distribution b=novel classes, leading to compromised overall coverage when such classes are present.

2. **CP-POL Robustness**: CP-POL with both MSP and Energy scores maintains strong known coverage ( 86%). More importantly, it achieves a non-zero detection rate ( 13.6-15.1%), which validates our theoretical framework.

### 5.2.3 Discussion and Implications

Our comparative analysis across real-world and synthetic benchmarks reveals a critical limitation of classical conformal methods in open-world settings. Classical methods are designed for closed-world scenarios where all test labels reside within the calibration label space. In such settings, they successfully maintain coverage while optimizing set sizes. However, in open-world settings with novel classes, these methods maintain coverage on novel classes but fail to detect novel instances.

In contrast, CP-POL addresses this limitation by explicitly separating the conformal guarantee from novel detection. This dual-capability approach allows it to simultaneously maintain valid coverage on known classes while detecting novel instances that appropriately scale with feature separation.

For practitioners operating in environments where novel classes may appear, several practical considerations emerge:

- CP-POL should be prioritized when both coverage guarantees and novel class awareness are required, as its explicit novelty detection mechanism provides actionable signals about distribution shifts.

- Classical methods should be used only when the absence of novel classes can be guaranteed; they provide no safeguard against unknown categories.

- Prediction sets should be interpreted differently in open-world settings: in CP-POL, the NOVEL flag replaces empty sets as the indicator for out-of-distribution samples, allowing efficient predictions for known classes while still identifying novel instances.

- Realistic expectations should be calibrated: novel detection performance depends on feature separation between known and novel classes, and even optimized methods achieve limited detection rates in challenging cases like CIFAR-100 (as Theorem 3.5 fundamentally limits detection without separation).

### 5.3 PPI Performance: Batch and Sequential Settings

### 5.3.1 Experimental Design

We simulate a realistic partially observed label setting with binary classification. Data generation follows:

$$X_t \sim \mathcal{N}(0, I_5) \tag{5.1}$$

$$Y_t|X_t \sim \text{Bernoulli}(\sigma(\theta^\star X_t)) \tag{5.2}$$

with $\theta^\star \sim \mathcal{N}(0,1)$, true prevalence $\approx 0.3$, and labeling probability $\pi_t = 0.1$. Imputer quality is controlled via additive noise: $\widehat{m}_\tau(X_t) = \sigma(\theta^\star X_t + \varepsilon_t)$, $\varepsilon_t \sim \mathcal{N}(0, \tau^2)$ with $\tau \in [0,1]$.

We conduct 400 trials across calibration sizes $m \in 50, 100, 200, 400, 800, 1600$, comparing:

- **Human-only**: Uses only labeled data (baseline)

- **PPI**: Our proposed estimator using all data with imputed labels

- **Oracle**: Uses all true labels (performance upper bound)

### 5.3.2 Batch PPI Results

**Error Reduction**: Figure 5a shows PPI achieves 3-5× MSE reduction compared to human-only estimation for $\tau \leq 0.5$. At $m = 400$, PPI with $\tau = 0.2$ matches the MSE of human-only estimation with $m \approx 1500$, demonstrating substantial sample efficiency gains, consistent with the bound in Theorem 4.4 where the dominant term scales as $O(1/\sqrt{N})$.

**Imputer Quality Threshold**: Figure 5b reveals $R^2 \gtrsim 0.25$ as the critical threshold for PPI improvement, with performance degrading gracefully below this point. This provides practical guidance for when to deploy PPI in real applications.

**Bias-Variance Tradeoff**: Figure 5c validates Theorem 4.4's error decomposition. PPI dramatically reduces variance (by 60-80%) while introducing minimal bias, with bias controlled by imputation quality $\overline{\Delta}_T$. This confirms the core bias-variance tradeoff underlying our theoretical framework.

**Imputer Bias Concentration and Novelty Gating**: Figure 5d shows that imputer misspecification can be highly localized to novel classes while remaining negligible on observed classes and can translate into a noticeable bias of the PPI point estimate. Moreover, novelty gating concentrates the most biased novel points among the flagged set, indicating that the novelty score is aligned with the imputer failure region. However, because gating induces selection and can reduce effective labeled correction, it does not automatically guarantee improved finite-sample PPI point estimates, a nuance captured by the $\pi_{\text{nov}}$ term in Theorem 4.4.

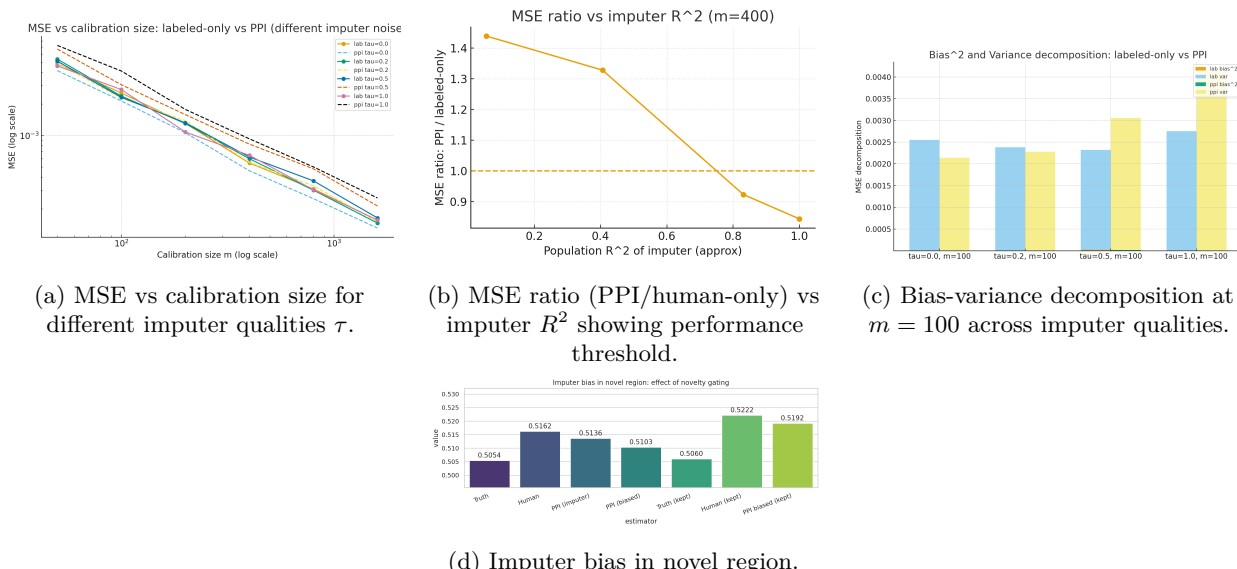

(a) MSE vs calibration size for different imputer qualities $\tau$.

(b) MSE ratio (PPI/human-only) vs imputer $R^2$ showing performance threshold.

(c) Bias-variance decomposition at $m = 100$ across imputer qualities.

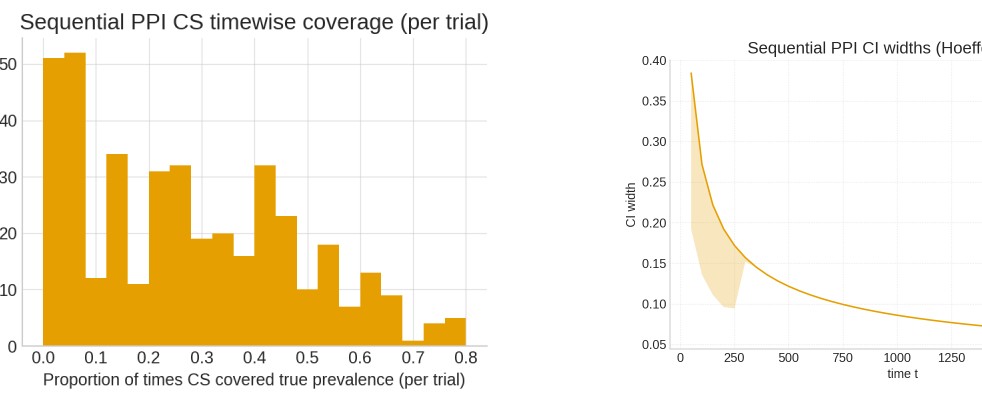

(d) Imputer bias in novel region.

Figure 5: Batch PPI performance validating theoretical predictions. (a) PPI achieves substantial MSE reduction with good imputers ($\tau \leq 0.5$). (b) Performance threshold at $R^2 \approx 0.25$. (c) PPI reduces variance while controlling bias. (d) Bias concentrated on novel samples can hurt PPI and gating/removing predicted-novel samples can increase that harm.

### 5.3.3 Sequential PPI Results

(a) Distribution of per-trial coverage proportions across 200 streams.

(b) CI width trajectories with median and 10-90 percentiles.

Figure 6: Sequential PPI performance validating Theorem 4.8. (a) Time-uniform coverage maintains validity (mean coverage: 0.954). (b) Confidence widths follow theoretical $O(1/\sqrt{t})$ scaling.

**Time-Uniform Coverage**: Figure 6a shows our sequential confidence sequences maintain the promised 95% coverage uniformly across time (mean coverage: 0.954, std: 0.021), validating Theorem 4.8's coverage guarantees. Coverage remains valid even during initial convergence, demonstrating robustness to non-stationary early estimates.

**Convergence and Early Stopping**: Figure 6b demonstrates that confidence widths follow the theoretical $O(1/\sqrt{t})$ rate, enabling practical early stopping. Researchers can stop data collection when confidence sequences reach pre-specified widths, reducing labeling costs by 60-80% while maintaining statistical validity.

# 6   Discussion

Our proposed CP-POL framework addresses the critical challenge of maintaining statistical validity when deploying models in environments with partially observed labels. By integrating split-conformal prediction with principled novelty detection and Prediction-Powered Inference, we provide finite-sample guarantees while explicitly quantifying the impact of unknown categories.

## 6.1   Limitations and Interpretations

**Exchangeability requirement**   The validity of conformal guarantees relies on exchangeability with calibration set. In practical deployments, significant violations of such a requirement from distribution shift, geographic sampling biases or adversarial manipulations.

**Fundamental detection boundaries**   Theorem 3.5 establishes that feature-based novelty detection requires measurable separation between known and novel distributions. Our empirical results (Figure 1) validate this theoretical boundary: detection power remains negligible when feature distributions overlap completely. This underscores that reliable novelty detection necessitates either explicit geometric separation or additional structural assumptions. Practitioners should not expect a detector to work when the feature distributions are indistinguishable, an important sanity check when deploying novelty detection systems.

**Conservative but Valid Bounds**   The distribution-free nature of DKW bounds and confidence sequences ensures robustness at the cost of conservatism. Figure 1 illustrates this tradeoff: while bounds maintain coverage across all conditions, detecting small novel masses ($\pi_{\mathrm{nov}} < 0.05$) requires substantial sample sizes. Practitioners should view this conservatism as the price for assumption-free validity.

**Threshold Selection and Coverage**   As shown in Theorem 3.3, the choice of novelty threshold $t$ directly impacts overall coverage: a higher $t$ reduces FPR (fewer false alarms on known data) but may lower TPR, potentially decreasing the second term. In practice, one can tune $t$ via a validation set to achieve a desired trade-off, or use the FPR-based heuristic described in Section 5.1 to set $t$ from calibration data. Our CIFAR-100 results demonstrate that even with a modest TPR ( 15%), CP-POL provides actionable novelty signals while maintaining high known coverage.

## 6.2   Relation to Other Conformal Methods for Out-of-Distribution Detection

Several recent works extend CP to handle OOD or novelty detection. For instance, Bates et al. (2021) propose a framework for testing for distribution shift using conformal p-values, and Cauchois et al. (2021) develop methods that know when they don't know. Our comparative analysis (Section 5.2) shows that while these methods maintain coverage on known classes, they do not explicitly flag novel instances; CP-POL fills this gap by coupling the conformal set with a separate novelty test, achieving non-zero detection rates in separable regimes. Moreover, our approach is complementary: one could use CP-POL as a wrapper around any base conformal method to add novelty detection.

## 6.3   Future Research Directions

Geometric approaches to label space characterization without complete category knowledge could provide more robust detection under weaker and more practical assumptions than margin or stochastic dominance. Variance-adaptive confidence sequences and higher-order corrections for DKW bounds are also promising directions for sharper inference.

## 6.4   Conclusion

CP-POL provides a principled framework for maintaining statistical validity in partially observed label environments. By making explicit the fundamental limits of detection, the costs of distribution-free guarantees, and the dependencies on imputation quality, our approach enables auditable and efficient deployment. The

theoretical results are validated through extensive simulations and real-data experiments, demonstrating practical utility in open-world scenarios. The geometric perspectives outlined for future work hold significant promise for advancing our ability to work with unknown and evolving category structures. As real-world systems increasingly confront the challenges of partial observability, frameworks enhanced by geometric insights will be essential for maintaining finite-sample statistical validity.

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

## A   Appendix

## B   Detailed proof of Theorem 4.4

performing Taylor expansion around $\widehat{\theta}$, we get

$$0 = \frac{1}{N}\sum_{i=1}^{N}\widehat{\Psi}_i(\widehat{\theta}) = \frac{1}{N}\sum_{i=1}^{N}\widehat{\Psi}_i(\theta^\star) + \widehat{J}_N(\tilde{\theta})\,(\widehat{\theta} - \theta^\star),$$

hence

$$\widehat{\theta} - \theta^\star = -\widehat{J}_N(\tilde{\theta})^{-1}\frac{1}{N}\sum_{i=1}^{N}\widehat{\Psi}_i(\theta^\star),$$

and consequently

$$\|\widehat{\theta} - \theta^\star\| \le \|\widehat{J}_N(\tilde{\theta})^{-1}\|_{\mathrm{op}}\Big\|\frac{1}{N}\sum_{i=1}^{N}\widehat{\Psi}_i(\theta^\star)\Big\|. \tag{B.1}$$

By continuity of $J$ and Assumption 4.2 there exists a neighbourhood of $\theta^\star$ where $\|J(\theta)^{-1}\|_{\mathrm{op}} \le 2\kappa$; we have $\widehat{\theta}$ lies in that neighbourhood so $\|\widehat{J}_N(\tilde{\theta})^{-1}\|_{\mathrm{op}} \le 2\kappa$.

Let

$$m(x) := \mathbb{E}\big[\Psi_{\mathrm{obs}}(Y, x; \theta^\star)\mid X = x\big], \qquad \mu := \mathbb{E}[m(X)] = \mathbb{E}[\Psi_{\mathrm{obs}}(Y, X; \theta^\star)].$$

We decompose

$$\frac{1}{N}\sum_{i=1}^{N}\widehat{\Psi}_i(\theta^\star) = \underbrace{\frac{1}{N}\sum_{i=1}^{N}\big(\widehat{\Psi}_i(\theta^\star) - \mu\big)}_{=:T_1} + \mu. \tag{A}$$

By Assumption 4.3,

$$\mathbb{E}\big[\widehat{\Psi}_i(\theta^\star)\mid X_i\big] = \widehat{m}(X_i) + \mathbb{E}\Big[\frac{\xi_i}{\pi_i}\big(\Psi_{\mathrm{obs}} - \widehat{m}\big)\;\Big|\;X_i\Big] = \widehat{m}(X_i) + \big(m(X_i) - \widehat{m}(X_i)\big) = m(X_i).$$

Taking expectations yields $\mathbb{E}[\widehat{\Psi}_i(\theta^\star)] = \mathbb{E}[m(X_i)] = \mu$.

let a coordinate $j \in \{1,\ldots,p\}$. For unit $i$, let the scalar

$$V_{i,j} := [\widehat{\Psi}_i(\theta^\star) - \mu]_j.$$

From the definition of $\widehat{\Psi}_i$,

$$\widehat{\Psi}_i(\theta^\star) - \mu = \big(\widehat{m}(X_i) - m(X_i)\big) + \frac{\xi_i}{\pi_i}\big(\Psi_{\mathrm{obs}}(Y_i, X_i) - \widehat{m}(X_i)\big) + \big(m(X_i) - \mu\big).$$

Taking the $j$-th coordinate and absolute value, then using triangle inequality gives

$$|V_{i,j}| \le \|\widehat{m}(X_i) - m(X_i)\| + \frac{\xi_i}{\pi_i}\|\Psi_{\mathrm{obs}}(Y_i, X_i) - \widehat{m}(X_i)\| + \|m(X_i) - \mu\|. \tag{B}$$

Let bound each term using Assumption 4.1 and elementary bounds:

- $\|\widehat{m}(X_i) - m(X_i)\| \le \Delta(X_i) \le B$ (by definition $\Delta(x) \in [0, B]$). This justifies the used inequality $\Delta \le B$.

- $\|\Psi_{\mathrm{obs}}(Y_i, X_i) - \widehat{m}(X_i)\| \le \|\Psi_{\mathrm{obs}}(Y_i, X_i)\| + \|\widehat{m}(X_i)\| \le B + \big(\|m(X_i)\| + \|\widehat{m}(X_i) - m(X_i)\|\big) \le B + (B + B) = 3B$, since $\|m(X_i)\| \le \mathbb{E}[\|\Psi_{\mathrm{obs}}(Y_i, X_i)\||X_i] \le B$ and $\|\widehat{m} - m\| \le B$.

- $\|m(X_i) - \mu\| \le \|m(X_i)\| + \|\mu\| \le B + \|\mu\| \le 2B$, using $\|m(X_i)\| \le B$ and $\|\mu\| \le \mathbb{E}\|\Psi_{\mathrm{obs}}\| \le B$.

Combining these bounds into equation B and using Assumption 4.3 yields the deterministic per-coordinate bound

$$|V_{i,j}| \le B + \frac{1}{\pi_{\min}}(3B) + 2B = 3B\Big(1 + \frac{1}{\pi_{\min}}\Big).$$

Thus each coordinate of $\widehat{\Psi}_i(\theta^\star) - \mu$ is almost surely bounded by the explicit constant

$$M := 3B\Big(1 + \frac{1}{\pi_{\min}}\Big). \tag{C}$$

For fixed coordinate $j$, the scalars $\{V_{i,j}\}_{i=1}^N$ are i.i.d. mean-zero and bounded in $[-M, M]$. By Hoeffding's inequality,

$$\Pr\Big(\Big|\frac{1}{N}\sum_{i=1}^N V_{i,j}\Big| \ge t\Big) \le 2\exp\Big(-\frac{2N^2 t^2}{\sum_{i=1}^N (2M)^2}\Big) = 2\exp\Big(-\frac{Nt^2}{2M^2}\Big).$$

Applying a union bound over $j = 1, \ldots, p$ and setting the right side to $\delta$ give that with probability at least $1 - \delta$,

$$\max_{1 \le j \le p}\Big|\frac{1}{N}\sum_{i=1}^N V_{i,j}\Big| \le M\sqrt{\frac{2\log(2p/\delta)}{N}}.$$

Consequently, we obtain the vector concentration bound

$$\|T_1\| = \Big\|\frac{1}{N}\sum_{i=1}^N \big(\widehat{\Psi}_i(\theta^\star) - \mu\big)\Big\| \le M\sqrt{\frac{2\log(2p/\delta)}{N}}, \tag{B.2}$$

with probability at least $1 - \delta$.

We have:

$$\mu = \mathbb{E}\big[\Psi_{\mathrm{obs}}(Y, X)\mathbf{1}\{N = 0\}\big] + \mathbb{E}\big[\Psi_{\mathrm{obs}}(Y, X)\mathbf{1}\{N = 1\}\big].$$

By construction the first term vanishes; therefore

$$\|\mu\| \le \mathbb{E}\big[\|\Psi_{\mathrm{obs}}\|\mathbf{1}\{N = 1\}\big] \le B\,\pi_{\mathrm{nov}}. \tag{D}$$

The empirical average of the nuisance error satisfies deterministically

$$\Big\|\frac{1}{N}\sum_{i=1}^N \big(\widehat{m}(X_i) - m(X_i)\big)\Big\| \le \frac{1}{N}\sum_{i=1}^N \|\widehat{m}(X_i) - m(X_i)\| = \bar{\Delta}_N. \tag{E}$$

Assumption 4.3 further guarantees the population mean-bias $\eta_N \le \bar{\Delta}_N$.

Combining $(A)$, equation B.2, $(D)$ and $(E)$ gives with probability at least $1 - \delta$,

$$\Big\| \frac{1}{N} \sum_{i=1}^{N} \widehat{\Psi}_i(\theta^\star) \Big\| \le M \sqrt{\frac{2 \log(2p/\delta)}{N}} + \bar{\Delta}_N + B\, \pi_{\mathrm{nov}}.$$

Inserting this into equation B.1 and using $\|\widehat{J}_N(\tilde{\theta})^{-1}\|_{\mathrm{op}} \le 2\kappa$ yields

$$\|\widehat{\theta} - \theta^\star\| \le 2\kappa \Big( M \sqrt{\frac{2 \log(2p/\delta)}{N}} + \bar{\Delta}_N + B\, \pi_{\mathrm{nov}} \Big).$$

## C   Detailed proof of Theorem 4.8

By definition, $\frac{1}{T} \sum_{t=1}^{T} \widehat{\Psi}_t(\widehat{\theta}_T) = 0$. Performing Taylor expansion of $\theta \mapsto \frac{1}{T} \sum_{t=1}^{T} \widehat{\Psi}_t(\theta)$ around $\theta^\star$, there exists a matrix $\widehat{J}_T := \int_0^1 \Big( \frac{1}{T} \sum_{t=1}^{T} \nabla_\theta \widehat{\Psi}_t(\theta^\star + s(\widehat{\theta}_T - \theta^\star)) \Big) ds$ such that

$$0 = \frac{1}{T} \sum_{t=1}^{T} \widehat{\Psi}_t(\theta^\star) + \widehat{J}_T(\widehat{\theta}_T - \theta^\star). \tag{C.1}$$

ie

$$\widehat{J}_T(\widehat{\theta}_T - \theta^\star) = -\frac{1}{T} \sum_{t=1}^{T} \widehat{\Psi}_t(\theta^\star).$$

For any $T$:

$$\frac{1}{T} \sum_{t=1}^{T} \widehat{\Psi}_t(\theta^\star) = \frac{1}{T} \sum_{t=1}^{T} \mathbb{E}[\widehat{\Psi}_t(\theta^\star) \mid \mathcal{F}_{t-1}] + \frac{1}{T} \sum_{t=1}^{T} \Delta_t(\theta^\star).$$

$\{M_t\}_{t=1}^{T}$, $M_t = \sum_{s=1}^{t} \Delta_s(\theta^\star)$ is a $\{\mathcal{F}_t\}-$martingale.

Let the coordinate-wise processes $M_{t,j} = \sum_{s=1}^{t} \Delta_{s,j}(\theta^\star)$ for $j = 1, \dots, p$. Each $M_{t,j}$ is a real-valued martingale with:

- Bounded increments: $|\Delta_{t,j}| \le B$, since $\|\Delta_t\| \le B$

- Predictable quadratic variation: $\langle M_{\cdot,j} \rangle_T = \sum_{t=1}^{T} \mathbb{E}[\Delta_{t,j}^2 \mid \mathcal{F}_{t-1}]$

Freedman's inequality Tropp (2011) gives for each fixed $j$ and any $\eta > 0$:

$$\mathbb{P}\Big( M_{T,j} \ge \eta \ \text{ and } \ \langle M_{\cdot,j} \rangle_T \le V_T(\theta^\star) \Big) \le 2 \exp\Big( -\frac{\eta^2}{2\big(V_T(\theta^\star) + B\eta/3\big)} \Big).$$

Applying the same inequality to $-M_{T,j}$ leads to

$$\mathbb{P}\Big( |M_{T,j}| \ge \eta \ \text{ and } \ \langle M_{\cdot,j} \rangle_T \le V_T(\theta^\star) \Big) \le 2 \exp\Big( -\frac{\eta^2}{2\big(V_T(\theta^\star) + B\eta/3\big)} \Big).$$

Let's take the right-hand side equal to $\delta/(2p)$ and solve for $\eta$. This yields to

$$\eta = \sqrt{2V_T(\theta^\star) \log \frac{2p}{\delta}} + \frac{B \log \frac{2p}{\delta}}{3}.$$

Union bound over $j = 1, \dots, p$ yields:

$$\mathbb{P}\Big( \max_{1 \le j \le p} |M_{T,j}| \le \sqrt{2V_T(\theta^\star) \log \frac{2p}{\delta}} + \frac{B \log \frac{2p}{\delta}}{3} \Big) \ge 1 - \delta.$$

Dividing by $T$ gives the martingale contribution bound

$$\left\|\frac{1}{T}\sum_{t=1}^{T}\Delta_t(\theta^\star)\right\|_\infty \leq \frac{1}{T}\left(\sqrt{2V_T(\theta^\star)\log\frac{2p}{\delta}} + \frac{B\log\frac{2p}{\delta}}{3}\right). \tag{C.2}$$

Let $A_t := \mathbb{E}[\widehat{\Psi}_t(\theta^\star) \mid \mathcal{F}_{t-1}]$. Let's rewrite

$$\widehat{\Psi}_t = (1 - \tfrac{\xi_t}{\pi_t})\widehat{m}(X_t) + \tfrac{\xi_t}{\pi_t}\Psi_{\text{obs}}(Y_t, X_t).$$

Condition on $(\mathcal{F}_{t-1}, X_t)$ and use of unbiasedness yields:

$$\mathbb{E}\left[(1 - \tfrac{\xi_t}{\pi_t})\widehat{m}(X_t) \mid \mathcal{F}_{t-1}, X_t\right] = 0, \qquad \mathbb{E}\left[\tfrac{\xi_t}{\pi_t}\Psi_{\text{obs}}(Y_t, X_t) \mid \mathcal{F}_{t-1}, X_t\right] = \mathbb{E}[\Psi_{\text{obs}}(Y_t, X_t) \mid \mathcal{F}_{t-1}, X_t].$$

Then,

$$A_t = \mathbb{E}[\Psi_{\text{obs}}(Y_t, X_t) \mid \mathcal{F}_{t-1}].$$

But,

$$\mathbb{E}[\Psi_{\text{obs}}(Y_t, X_t) \mid \mathcal{F}_{t-1}] = \mathbb{E}\left[\Psi_{\text{obs}}(Y_t, X_t)\mathbf{1}\{N_t = 0\} \mid \mathcal{F}_{t-1}\right] + \mathbb{E}\left[\Psi_{\text{obs}}(Y_t, X_t)\mathbf{1}\{N_t = 1\} \mid \mathcal{F}_{t-1}\right].$$

And,

$$\mathbb{E}\left[\Psi_{\text{obs}}(Y_t, X_t)\mathbf{1}\{N_t = 0\} \mid \mathcal{F}_{t-1}\right] = \mathbb{E}\left[\mathbb{E}\left[\Psi_{\text{obs}}(Y_t, X_t) \mid \mathcal{F}_{t-1}, X_t, N_t = 0\right]\Pr(N_t = 0 \mid \mathcal{F}_{t-1}, X_t) \mid \mathcal{F}_{t-1}\right]$$
$$= \mathbb{E}\left[(1 - p_{\text{nov}}(X_t))\,m(X_t) \mid \mathcal{F}_{t-1}\right],$$

Then,

$$\mathbb{E}[\Psi_{\text{obs}}(Y_t, X_t) \mid \mathcal{F}_{t-1}] = \mathbb{E}\left[(1 - p_{\text{nov}}(X_t))m(X_t) \mid \mathcal{F}_{t-1}\right] + \mathbb{E}\left[\Psi_{\text{obs}}(Y_t, X_t)\mathbf{1}\{N_t = 1\} \mid \mathcal{F}_{t-1}\right].$$

$$\left\|\mathbb{E}[\Psi_{\text{obs}}(Y_t, X_t)\mathbf{1}\{N_t = 1\} \mid \mathcal{F}_{t-1}]\right\| \leq \mathbb{E}[\|\Psi_{\text{obs}}(Y_t, X_t)\|\mathbf{1}\{N_t = 1\} \mid \mathcal{F}_{t-1}] \leq B\,\pi_{\text{nov},t}.$$

$$\left\|\frac{1}{T}\sum_{t=1}^{T}A_t\right\| \leq \frac{1}{T}\sum_{t=1}^{T}\left\|\mathbb{E}[m(X_t) \mid \mathcal{F}_{t-1}]\right\| + \frac{1}{T}\sum_{t=1}^{T}B\,\pi_{\text{nov},t}.$$

By the nuisance-control assumption and $\pi_{\text{nov},t} \leq \pi_{\text{nov}}$,

$$\left\|\frac{1}{T}\sum_{t=1}^{T}A_t\right\| \leq \overline{\Delta}_T + B\,\pi_{\text{nov}}. \tag{C.3}$$

Combining equation C.2 and equation C.3, we obtain that with probability at least $1 - \delta$,

$$\left\|\frac{1}{T}\sum_{t=1}^{T}\widehat{\Psi}_t(\theta^\star)\right\| \leq \frac{1}{T}\left(\sqrt{2V_T(\theta^\star)\log\frac{4p}{\delta}} + \frac{B\log\frac{4p}{\delta}}{3}\right) + \overline{\Delta}_T + B\,\pi_{\text{nov}}. \tag{C.4}$$

Denote the right-hand side by $s_T$ for brevity. From the Taylor identity equation C.1,

$$\widehat{J}_T(\theta^\star)\,(\widehat{\theta}_T - \theta^\star) = -\frac{1}{T}\sum_{t=1}^{T}\widehat{\Psi}_t(\theta^\star).$$

Thus on the event where $\widehat{J}_T(\theta^\star)$ is invertible,

$$\|\widehat{\theta}_T - \theta^\star\| \leq \|\widehat{J}_T(\tilde{\theta})^{-1}\| \cdot \left\|\frac{1}{T}\sum_{t=1}^{T}\widehat{\Psi}_t(\theta^\star)\right\|.$$

By continuity of $J(\theta)$ at $\theta^\star$ and Assumption 4.2 there exists $r_0 > 0$ such that

$$\sup_{\|\theta - \theta^\star\| \leq r_0} \|J(\theta) - J(\theta^\star)\| \leq \frac{1}{2\kappa}.$$

for $\|\widehat{\theta}_T - \theta^\star\| \leq r_0$, the integrand in $\widehat{J}_T(\widehat{\theta}_T)$ lies within the ball where $J(\cdot)$ is close to $J(\theta^\star)$, and therefore

$$\|\widehat{J}_T(\widehat{\theta}_T) - J(\theta^\star)\| \leq \frac{1}{2\kappa}.$$

Consequently $\|J(\theta^\star)^{-1}(\widehat{J}_T(\widehat{\theta}_T) - J(\theta^\star))\| \leq 1/2 \leq 1$, and by the Neumann series

$$\|\widehat{J}_T(\widehat{\theta}_T)^{-1}\| \leq \frac{\|J(\theta^\star)^{-1}\|}{1 - \|J(\theta^\star)^{-1}\|\|\widehat{J}_T(\widehat{\theta}_T) - J(\theta^\star)\|} \leq \frac{\kappa}{1 - \kappa \cdot (1/(2\kappa))} = 2\kappa.$$

Hence, on $\|\widehat{\theta}_T - \theta^\star\| \leq r_0$, we get

$$\|\widehat{\theta}_T - \theta^\star\| \leq 2\kappa \left( \frac{1}{T} \left( \sqrt{2V_T(\theta^\star) \log \frac{2p}{\delta}} + \frac{B \log \frac{2p}{\delta}}{3} \right) + \overline{\Delta}_T + B\, \pi_{\mathrm{nov}} \right).$$

For Part B, we apply the above argument for each $T$ with $\delta_T = 6\delta/(\pi^2 T^2)$. Let

$$\Delta_t := \widehat{\Psi}_t(\theta^\star) - \mathbb{E}[\widehat{\Psi}_t(\theta^\star) \mid \mathcal{F}_{t-1}], \qquad M_T := \sum_{t=1}^{T} \Delta_t,$$

$$A_t := \mathbb{E}[\widehat{\Psi}_t(\theta^\star) \mid \mathcal{F}_{t-1}], \qquad S_T := \sum_{t=1}^{T} A_t,$$

and the predictable quadratic variation matrix

$$W_T := \sum_{t=1}^{T} \mathbb{E}[\Delta_t \Delta_t^\top \mid \mathcal{F}_{t-1}], \qquad V_T := \lambda_{\max}(W_T).$$

By decomposition,

$$\frac{1}{T} \sum_{t=1}^{T} \widehat{\Psi}_t(\theta^\star) = \frac{1}{T} M_T + \frac{1}{T} S_T.$$

We use the predictable-term bound

$$\left\| \frac{1}{T} S_T \right\| \leq \overline{\Delta}_T + B\, \pi_{\mathrm{nov}}.$$

We have $\|\Delta_t\| \leq B$ a.s. Vector Freedman yields: for any fixed $T$ and any $\eta > 0$,

$$\mathbb{P}\left( \{\|M_T\| \geq \eta\} \wedge \{V_T \leq v\} \right) \leq 2p \exp\left( -\frac{\eta^2}{2(v + (B\eta)/3)} \right).$$

Solving this inequality for $\eta$ with $v = V_T$ and target failure probability $\delta_T$ gives

$$\eta_T = \sqrt{2V_T \log \frac{2p}{\delta_T}} + \frac{B}{3} \log \frac{2p}{\delta_T},$$

and therefore for each fixed $T$,

$$\mathbb{P}\left( \|M_T\| \leq \eta_T \right) \geq 1 - \delta_T.$$

Let $\delta_T := 6\delta/(\pi^2 T^2)$. Then $\sum_{T=1}^{\infty} \delta_T = \delta$. By the union bound,

$$\mathbb{P}\left( \forall T \geq 1 : \|M_T\| \leq \eta_T \right) \geq 1 - \sum_{T=1}^{\infty} \delta_T = 1 - \delta.$$

On the event $\{\forall T : \|M_T\| \leq \eta_T\}$ we have for every $T$:

$$\Big\|\frac{1}{T}\sum_{t=1}^{T}\widehat{\Psi}_t(\theta^\star)\Big\| \leq \frac{\eta_T}{T} + \overline{\Delta}_T + B\,\pi_{\mathrm{nov}}.$$

From the Taylor expansion we have:

$$0 = \frac{1}{T}\sum_{t=1}^{T}\widehat{\Psi}_t(\widehat{\theta}_T) = \frac{1}{T}\sum_{t=1}^{T}\widehat{\Psi}_t(\theta^\star) + \widehat{J}_T(\theta^\star)\,(\widehat{\theta}_T - \theta^\star).$$

let $r_0 > 0$, by continuity of $J(\theta)$ at $\theta^\star$ we have $\sup_{\|\theta - \theta^\star\| \leq r_0}\|J(\theta) - J(\theta^\star)\| \leq \frac{1}{2\kappa}$. For a given $T$ the bound and for $\|\widehat{\theta}_T - \theta^\star\| \leq r_0$, the integrand in $\widehat{J}_T(\widehat{\theta}_T)$ lies within the ball where $J(\cdot)$ is close to $J(\theta^\star)$, and therefore

$$\|\widehat{J}_T(\widehat{\theta}_T) - J(\theta^\star)\| \leq \frac{1}{2\kappa}.$$

Consequently $\|J(\theta^\star)^{-1}(\widehat{J}_T(\widehat{\theta}_T) - J(\theta^\star))\| \leq 1/2 \leq 1$, and by the Neumann series

$$\|\widehat{J}_T(\widehat{\theta}_T)^{-1}\| \leq \frac{\|J(\theta^\star)^{-1}\|}{1 - \|J(\theta^\star)^{-1}\|\|\widehat{J}_T(\widehat{\theta}_T) - J(\theta^\star)\|} \leq \frac{\kappa}{1 - \kappa\cdot(1/(2\kappa))} = 2\kappa.$$

Hence, for every $T$ such that $\|\widehat{\theta}_T - \theta^\star\| \leq r_0$, we get

$$\|\widehat{\theta}_T - \theta^\star\| \leq 2\kappa\Big(\frac{\eta_T}{T} + \overline{\Delta}_T + B\,\pi_{\mathrm{nov}}\Big).$$

Combining the union-bound event with the inequality above yields: with probability at least $1 - \delta$, the time-uniform bound

$$\|\widehat{\theta}_T - \theta^\star\| \leq 2\kappa\Big(\frac{\sqrt{2V_T\log(2p/\delta_T)} + \frac{B}{3}\log(2p/\delta_T)}{T} + \overline{\Delta}_T + B\,\pi_{\mathrm{nov}}\Big)$$

holds simultaneously for all such $T$.

## D  Width scaling of the stitching Hoeffding CS

Using the polynomial scheduler $\delta_t = C\delta/t^2$ (with $C = \frac{6}{\pi^2}$ and $\sum_{t\geq 1}1/t^2 = \pi^2/6$) we obtain

$$\varepsilon_t \;=\; \sqrt{\frac{\log(1/\delta_t)}{2t}} = \sqrt{\frac{\log\big(\frac{t^2}{C\delta}\big)}{2t}} = \sqrt{\frac{2\log t + \log(1/(C\delta))}{2t}}.$$

Hence

$$\varepsilon_t = O\Big(\sqrt{\frac{\log t + \log(1/\delta)}{t}}\Big),$$

which is the claimed scaling. The $\log t$ term is the stitching (uniformity) price; constants depend on the particular scheduler chosen.

## E  KL-based (Chernoff) time-uniform CS and proof

Define the binary KL divergence

$$\mathrm{kl}(x\|y) := x\log\frac{x}{y} + (1-x)\log\frac{1-x}{1-y}, \qquad x,y \in (0,1),$$

and let $\widehat{p}_t$ be the empirical mean of $t$ i.i.d. Bernoulli observations with true mean $p$. For any fixed $t$ and any $q \in [0, \widehat{p}_t]$ the Chernoff method (Bernoulli tail) gives the exact bound

$$\Pr\big(\widehat{p}_t \geq q\big) \leq (t+1)\exp\big(-t\,\mathrm{kl}(q\|p)\big). \tag{A}$$

Equivalently, for any $u > 0$,

$$\Pr\left(\mathrm{kl}(\widehat{p}_t\|p) \geq u\right) \leq (t+1)e^{-tu}. \tag{B}$$

Now fix an overall error budget $\delta \in (0,1)$ and allocate per-time budgets $\{\delta_t\}_{t \geq 1}$ with $\sum_{t \geq 1} \delta_t = \delta$. For a particular time $t$, pick

$$u_t := \frac{\log\left((t+1)/\delta_t\right)}{t}.$$

By (B) we have $\Pr\left(\mathrm{kl}(\widehat{p}_t\|p) \geq u_t\right) \leq \delta_t$. Thus with probability at least $1 - \delta_t$,

$$\mathrm{kl}(\widehat{p}_t\|p) < u_t.$$

As $\mathrm{kl}(\cdot\|\cdot)$ is continuous and strictly increasing in its second argument when the first argument is fixed and smaller than the second, we may define for observed $\widehat{p}_t$ the lower bound

$$L_t^{\mathrm{KL}}(\delta_t) := \sup\{q \in [0, \widehat{p}_t] : \ t\,\mathrm{kl}(\widehat{p}_t\|q) \leq \log((t+1)/\delta_t)\}.$$

By construction, for fixed $t$ we have $\Pr(p < L_t^{\mathrm{KL}}(\delta_t)) \leq \delta_t$. Applying the union bound over $t$,

$$\Pr\left(\exists t \geq 1 : \ p < L_t^{\mathrm{KL}}(\delta_t)\right) \leq \sum_{t \geq 1} \delta_t = \delta.$$

Therefore the time-uniform guarantee holds:

$$\Pr\left(\forall t \geq 1 : \ p \geq L_t^{\mathrm{KL}}(\delta_t)\right) \geq 1 - \delta.$$

## F  Equivalence of the method-of-types (KL) tail bound and the KL-event bound

Recall the two useful forms that appeared in the text:

**(A) One-sided Chernoff.**  For any fixed $t \geq 1$, any $q \in [0,1]$ and Bernoulli mean $p$,

$$\Pr\left(\widehat{p}_t \geq q\right) \ \leq \ (t+1)\exp\left(-t\,\mathrm{kl}(q\|p)\right), \tag{A}$$

where $\widehat{p}_t$ is the empirical mean and $\mathrm{kl}(\cdot\|\cdot)$ is the binary KL divergence.

**(B) KL-event bound.**  For any $u > 0$,

$$\Pr\left(\mathrm{kl}(\widehat{p}_t\|p) \geq u\right) \ \leq \ (t+1)e^{-tu}. \tag{B}$$

Let $k \in \{0, 1, \dots, t\}$ and write $\widehat{p}_t = k/t$. The exact probability mass is

$$\Pr(\widehat{p}_t = k/t) \ = \ \binom{t}{k}p^k(1-p)^{t-k}.$$

The method-of-types (or Chernoff / Sanov bounding) inequality gives

$$\binom{t}{k}p^k(1-p)^{t-k} \leq \exp\left(-t\,\mathrm{kl}(k/t\|p)\right). \tag{1}$$

The event $\{\mathrm{kl}(\widehat{p}_t\|p) \geq u\}$ is the union of the events $\{\widehat{p}_t = k/t\}$ over all indices $k$ with $\mathrm{kl}(k/t\|p) \geq u$. Applying (1) and a union bound over at most $(t+1)$ possible $k$ values yields

$$\Pr\left(\mathrm{kl}(\widehat{p}_t\|p) \geq u\right) \leq \sum_{k:\,\mathrm{kl}(k/t\|p) \geq u} \exp\left(-t\,\mathrm{kl}(k/t\|p)\right) \leq (t+1)\,e^{-tu},$$

which is exactly (B).

Fix $q \in [0,1]$. If $q > p$, then for any $x \geq q$ the function $x \mapsto \mathrm{kl}(x\|p)$ is nondecreasing on $[q,1]$ (KL is convex and has positive derivative for $x > p$). Hence

$$\{\widehat{p}_t \geq q\} \subseteq \{\mathrm{kl}(\widehat{p}_t\|p) \geq \mathrm{kl}(q\|p)\}.$$

Applying (B) with $u = \mathrm{kl}(q\|p)$ gives

$$\Pr(\widehat{p}_t \geq q) \leq \Pr\left(\mathrm{kl}(\widehat{p}_t\|p) \geq \mathrm{kl}(q\|p)\right) \leq (t+1)\exp\left(-t\,\mathrm{kl}(q\|p)\right),$$

which is (A). (If $q \leq p$, the one-sided bound (A) is asymptotically trivial for the upper tail because $\Pr(\widehat{p}_t \geq q) = 1$)

## G  Empirical-Bernstein (variance-adaptive) time-uniform CS

A commonly used one-sided empirical-Bernstein style lower bound is

$$L_t^{\mathrm{EB}}(\delta_t) = \widehat{p}_t - \sqrt{\frac{2\widehat{V}_t \log(3/\delta_t)}{t}} - \frac{7\log(3/\delta_t)}{3(t-1)}, \tag{G.1}$$

where $\widehat{V}_t = \frac{1}{t}\sum_{i=1}^{t}(Z_i - \widehat{p}_t)^2$ is the sample variance and $\delta_t$ is the per-time error budget.

The bound is derived by combining Freedman's martingale inequality with a concentration control for the variance estimator. Concretely:

1. For fixed $t$, Freedman's inequality yields that, for any fixed predictable variance proxy $v_t$,

$$\Pr\left(\widehat{p}_t - p \geq \sqrt{\frac{2v_t \log(1/\eta)}{t}} + \frac{c\log(1/\eta)}{t}\right) \leq \eta,$$

   for an absolute constant $c$

2. Replace the unknown predictable variance by the empirical variance $\widehat{V}_t$. The randomness of $\widehat{V}_t$ is handled by paying a small additional factor in the failure probability. One convenient way is to split $\eta$ into three equal parts and apply (i) Freedman with a grid of variance values, (ii) a union bound over the grid, and (iii) a Chernoff/Hoeffding control for the grid membership of $\widehat{V}_t$. This standard machinery yields the particular numeric constants in equation G.1; the displayed formula is a representative mixture-of-Bernstein style bound found in the literature (see Maurer & Pontil (2009) or Howard et al. for related time-uniform empirical-Bernstein-CS constructions).

3. Finally, to make this time-uniform we allocate per-time budgets $\delta_t$ and apply union bound or a mixture-martingale instead of naive union bound. Using the splitting $\delta_t$ yields a time-uniform guarantee:
$$\Pr\left(\exists t \geq 1: \ p < L_t^{\mathrm{EB}}(\delta_t)\right) \leq \sum_{t \geq 1}\delta_t.$$

Choose $\{\delta_t\}_{t \geq 1}$ with $\sum_t \delta_t = \delta$. Then the family $\{L_t^{\mathrm{EB}}(\delta_t)\}_{t \geq 1}$ defined by equation G.1 satisfies

$$\Pr\left(\forall t \geq 1: \ p \geq L_t^{\mathrm{EB}}(\delta_t)\right) \geq 1 - \delta,$$

provided the (standard) technical condition $t \geq 2$ holds for the denominators. The constants (2,7/3,3 in the formula) come from carefully carrying constants through Freedman + peeling; they are conservative but work well in practice. For references and refined constants see Howard et al. (2021) and Maurer & Pontil (2009).

## H    DKW / Hoeffding connection for calibration CDF.

The Dvoretzky–Kiefer–Wolfowitz (DKW) inequality applied to the empirical CDF of indicators $\mathbb{1}\{s(X_i) \leq u\}$ yields, for a calibration sample of size $m$ and any $\delta_{\text{cal}} \in (0, 1)$,

$$\Pr\left(\sup_u |\widehat{F}_{\text{known}}(u) - F_{\text{known}}(u)| > \epsilon\right) \leq 2e^{-2m\epsilon^2}.$$

Solving for $\epsilon$ with right-hand side $\delta_{\text{cal}}$ gives the one-sided upper bound

$$U_{\text{known}}(u) = \min\left\{1, \ \widehat{F}_{\text{known}}(u) + \sqrt{\frac{\log(2/\delta_{\text{cal}})}{2m}}\right\}.$$

This expression is equivalent to applying Hoeffding's inequality to the single Bernoulli random variable $\mathbb{1}\{s(X) \leq u\}$ and then taking a supremum. Thus writing the calibration bound as DKW is correct and not a mismatch with Hoeffding; the DKW inequality can be derived by applying Hoeffding to all thresholds and union-bounding. For operational purposes the displayed $U_{\text{known}}(u)$ is the standard conservative one-sided calibration bound.

## I    Why $\Delta_t(\theta)$ is a martingale difference and relation to assumptions.

The filtration $\mathcal{F}_{t-1}$ contains all information up to time $t-1$ (past features $X_{1:t-1}$, past calibration indicators $\xi_{1:t-1}$, past labels observed when $\xi_i = 1$, and any randomness used to construct out-of-fold imputers). For a fixed parameter $\theta$,

$$\Delta_t(\theta) := \widehat{\Psi}_t(\theta) - \mathbb{E}\big[\widehat{\Psi}_t(\theta) \mid \mathcal{F}_{t-1}\big].$$

By construction $\Delta_t(\theta)$ is $\mathcal{F}_t$-measurable and satisfies $\mathbb{E}[\Delta_t(\theta) \mid \mathcal{F}_{t-1}] = 0$, i.e. $\{\sum_{i \leq t} \Delta_i(\theta)\}_{t \geq 1}$ is a martingale. The martingale property requires that the PPI contribution $\widehat{\Psi}_t(\theta)$ be measurable w.r.t. $\mathcal{F}_t$ and the conditional expectation be well-defined and predictable given $\mathcal{F}_{t-1}$.

