# OpenReview forum: "CP-POL + PPI: Conformal Guarantees in Partially-Observed Label Space"
_TMLR — Accepted by TMLR_

### Review · Reviewer_6D52 · 2026-01-12

**Summary Of Contributions:**

**Strengths**

1.The study innovatively integrates split conformal prediction, novelty detection, and prediction-powered inference, providing a unified framework with finite-sample guarantees and theoretical rigor for partially observed label spaces.

2.Through core theories such as the Le Cam impossibility result and structural separation conditions, it clearly defines the boundaries and feasibility of novelty detection, while simulation experiments effectively validate the method’s practicality in both batch and sequential settings.

**Weaknesses**

1.The paper’s Le Cam impossibility result in Theorem 3.5 and structural separation conditions rigorously define the limits of novelty detection, a key contribution. However, the practical verifiability of structural assumptions may be overlooked.

How can practitioners confirm uniform margin separation or stochastic dominance in real-world data where novel labels are unobserved?

The empirical validation uses simulated Beta distributions, but no guidance is provided for messy, high-dimensional real data, e.g., images, text,  where feature-novelty links are opaque.

2.The experiments rely on simulated data, including scalar novelty scores and binary classification, that simplify key real-world challenges. However, no evaluation on natural distribution shifts, the covariate shift between calibration and test data, which violate exchangeability, the core assumption of conformal guarantees.

Furthermore, similar to W1, novelty detection is tested on synthetic Beta distributions, but real-world novel labels often differ in semantic or structural ways, e.g., a new object class in computer vision, that may not align with scalar score separability.

Overall, this work is good. However, personally, the theoretical work should establish certain connections with the issues of practical concern or provide valuable perspectives.

**Additional Comments:**

Please refet to the Summary/Changes.

**Audience:**

No

**Audience Explanation:**

This paper is overly theoretical, with insufficient discussion on the data types and scenarios of practical concern, and may fail to attract the attention of relevant researchers.

**Claims And Evidence:**

Yes

**Claims Explanation:**

The theoretical derivations and simulation experiments have initially verified the arguments to be proved.

**Requested Changes:**

Please complete the discussions listed in the contributions.

---

> ### Author Response · Authors · 2026-01-20
> **Response to Reviewer 6D52**
>
> We thank the reviewer for the constructive and detailed review. We take your point seriously that the first version may read as overly theoretical and could better connect to practical concerns about verifiability, real data, and exchangeability violations.
>
> 1. On Practical verifiability of structural assumptions, we agree that margin separation and stochastic dominance are difficult to "verify" directly when novel labels are unobserved. Our contribution is to precisely delineate (i) when novelty detection is impossible without structure (via the Le Cam impossibility result), and (ii) what forms of structure suffice for guarantees. In the revision, we additionally clarify an operational strategy that does not require observing novel labels: practitioners can (a) set the novelty threshold by controlling the calibration false positive rate (FPR) on known data, and (b) use unlabeled pools to detect deviations in the score distribution and lower bound novel mass. Concretely, Sec. 5.1.2 now includes an explicit calibration-quantile rule $t := \widehat F_{\mathrm{known}}^{-1}(1-q)$ for user-chosen $q\in\{ 0.01,0.05,0.10 \}$ which makes the procedure deployable without access to novel labels. \
> We emphasize that this does not "prove" separation in a given application but provides a calibration-driven mechanism to operate safely and detect when the conditions necessary for meaningful novelty detection are not met which is exactly the content of the impossibility result.
> 2. On Real-world, high-dimensional data beyond scalar Beta simulations, we agree with the reviewer that synthetic scalar-score experiments alone are insufficient for practical assessment. To address this, we added a real-world withheld-classes experiment on CIFAR-100 (Sec. 5.1.3). In this experiment, a subset of classes is treated as observed during training/calibration, and the remaining classes are withheld and treated as novel at test time.
> 3. On exchangeability and distribution shift, we agree that exchangeability may be violated in practice due to covariate shift or adversarial effects. We make this limitation explicit in the Discussion under “Exchangeability requirement” (Sec. 6.1), clarifying that conformal validity relies on exchangeability with the calibration set and that violations can degrade guarantees. While the current revision does not add a full distribution-shift benchmark (which would broaden the scope significantly), we now explicitly communicate this assumption and motivate it as an important direction for future work.

---

> > ### Comment · Reviewer_6D52 · 2026-03-31
> > **Thanks for your response**
> >
> > I have read the authors' responses, yet remain not fully convinced, and the issues raised do exist. However, I fully understand that advancing relevant research in the field inevitably involves a progression from theory to application, and we encourage incremental exploration and experimentation. Overall, this work has made an attempt and provided some theoretical frameworks for the community.

---

### Review · Reviewer_fPN7 · 2026-01-12

**Summary Of Contributions:**

This paper addresses the limitation of classical Conformal Prediction (CP) in scenarios where training and calibration data observe only a subset of the full label space. It introduces CP-POL, a framework that combines Split CP with a calibrated novelty test and integrates Prediction-Powered Inference (PPI) for population estimation under these conditions. The authors provide a non-asymptotic theoretical framework including impossibility results and finite-sample bounds, which are validated through reproducible simulations.

The main contributions of this paper can be summarized as:
* This paper establishes a fundamental impossibility result using Le Cam's two-point method, proving that novelty detection from features alone is unidentifiable without structural assumptions.
* Identifies specific structural conditions (margin separation and stochastic dominance) that allow for valid coverage guarantees and novelty detection.
* Derives non-asymptotic, finite-sample lower bounds for the novel label mass ($\pi_{nov}$) using both batch (DKW-based) and sequential (martingale-based) approaches.
* Provides an algorithm named CP-POL, a practical pipeline for practitioners to deploy calibrated systems that can safely flag novel inputs rather than over-relying on model predictions

**Audience:**

Yes

**Audience Explanation:**

The paper intersects several active areas of machine learning research: Conformal Prediction, Open-Set Recognition, and Statistical Inference (PPI). Its focus on providing "auditable behavior" and "finite-sample guarantees" aligns with TMLR's interest in reliable and theoretically grounded machine learning.

**Claims And Evidence:**

Yes

**Claims Explanation:**

The proposed claims are supported by a combination of rigorous mathematical proofs and targeted simulations. The theoretical bounds (e.g., Theorems 3.8, 4.4, and 4.8) are derived with detailed proofs. The empirical results in Section 5 directly mirror the theoretical regimes: Figure 1 confirms the impossibility result versus the detectable regimes under separation, while Figures 2 and 3 demonstrate the MSE reduction and coverage validity of the PPI and sequential estimators. The evidence is clear and convincing within the scope of the presented simulations.

[Strengths]
* This paper addresses the "simple-but-hard reality" that deployed systems often encounter labels not present during training.
* The authors provide a comprehensive theoretical treatment from impossibility proofs to tight, non-asymptotic coverage bounds.
* The framework is applicable to both batch settings and streaming/sequential data environments through time-uniform confidence sequences.
* The paper clearly separates the fundamental limits of what can be known from the positive results achievable under realistic assumptions

[Weaknesses]
* Dependence on Structural Assumptions: The "positive results" for novelty detection rely on assumptions like margin separation, which may be difficult to verify or may not hold for complex, high-dimensional datasets
* Synthetic Evaluation: The empirical validation is restricted to simulated data; the lack of testing on real-world "long-tailed" or "open-set" datasets limits the immediate assessment of its practical performance.
* Threshold Sensitivity: While an optimal threshold ($t=0.7$) is identified in simulations, the paper provides limited guidance on how a practitioner should adaptively select this threshold in diverse real-world applications.

**Requested Changes:**

1. For experiments in Sec. 5, the paper's reliance on synthetic Beta-distributed scores may oversimplify real-world feature overlaps. It is suggested to include at least one experiment on a real-world dataset (e.g., ImageNet or a text classification task) where certain classes are withheld during calibration to simulate the partially-observed label space.
2. In Sec. 5.1.2, the novelty threshold $t$ is fixed. It is better to discuss or provide a heuristic for selecting $t$ based on a target False Positive Rate (FPR) on the calibration set, which would make the pipeline more operational for users.
3. While Fig. 2b discusses imputer $R^2$ thresholds, more detail on how CP-POL behaves when the imputer is biased in the "novel" region would be beneficial.

---

> ### Author Response · Authors · 2026-01-20
> **Response to Reviewer fPN7**
>
> We really Thank the reviewer for the constructive and detailed review.
>
> 1.We agree that Beta-distributed scalar scores oversimplify real-world overlaps. We therefore added a real-world withheld-classes experiment on CIFAR-100 to simulate the partially-observed label space. We report novelty-detection performance using negative MSP and Energy, and include a new figure summarizing score distributions and ROC behavior.\
> 2. On the guidance for choosing novelty threshold t, we added an explicit heuristic that selects t using a target calibration-set FPR. \
> 3. On the imputer bias in the "novel" region, we added a stress test where the imputer remains accurate on observed classes but is intentionally biased on withheld classes. We report that the induced misspecification is localized to the novel region and that novelty gating concentrates the most biased novel points among the flagged set. We also clarify a practical caveat: gating does not automatically guarantee finite-sample improvement of the PPI point estimate due to selection effects and reduced labeled correction.

---

### Review · Reviewer_EdpU · 2026-02-12

**Summary Of Contributions:**

This paper tackles the problem of preserving the coverage of conformal prediction's sets for observed classes while detecting when a sample at deployment does not belong to the observed classes ("novel"). The proposed approach is based on utilization of a novelty detector. Without distributional assumptions, a negative result is presented. Positive results are presented when building on distributional assumptions for defining the novelty detector (of limited practicality). Other extensions include deriving bounds on the novelty probability and integration with Prediction-Powered Inference (PPI) framework.
The empirical section is mostly based on simulations rather than real data.

**Audience:**

Yes

**Audience Explanation:**

This paper tackles a well motivated problem: preserving the coverage of conformal prediction's sets for observed classes while detecting when a sample at deployment does not belong to observed classes ("novel"). The negative result is useful. The positive results have their merits, though I am not sure about their practicality.

**Claims And Evidence:**

No

**Claims Explanation:**

The readability should be improved. There should also be more discussion on the relation to other CP methods that consider out-of-distribution / outlier detection. The experiment section should be improved as well.

See the comments below.

**Requested Changes:**

- The readability should be improved.  In particular, discuss more the implications of the theoretical results.
Remind the reader of notations when they are used.
Define P_{0,X} and P_{0,X} in Theorem 3.5.

- The conditions for the positive results in Section 3.1.3 do not seem to be verifiable (or hold) in practice when the distributions are unknown.

- The PPI Section 4 is hard to follow, and lacks discussions on the motivation, assumptions and results.
Where in the PPI paper is there a formula similar to Eq. 4.1?
Where in the PPI paper is it stated that "The PPI estimator θ_hat solves 1/N Psi_hat_i(θ_hat) = 0"?
Give preliminary details on PPI and carefully explain the relation of your results to existing PPI works, including explanations of your assumptions and those made in earlier works.

- Formally show the stated claim on the detectable regimes in Section 5.1.1.

- In Figure 1a, I do not see the "Dashed line shows theoretical detection boundary".
Discuss more how the results in Figure 1 corroborate the theory.

- There is no discussion or even pointer to Figure 2 (CIFAR experiments).

- The experiments do not explain well how to set the novelty detector and do not examine its effect on the coverage and other properties of CP.

- You should also discuss the relation and compare with other CP methods that consider out-of-distribution / outlier detection.

---

> ### Author Response · Authors · 2026-02-17
> **Response to Reviewer EdpU**
>
> We deeply Thank the reviewer for the constructive and detailed review.
>
> **Readability and Implications**:  We have restructured several parts of the paper to improve flow and clarity. In particular:
>
> * Added explicit definitions of notation, including $P_{0,X}$ and $P_{1,X}$ in Theorem 3.5.
>
> * Added a paragraph after the positive results (Section 3.1.3) discussing how practitioners can assess the plausibility of structural assumptions using calibration diagnostics (e.g., comparing empirical CDFs, using the DKW lower bound).
>
> * Expanded the PPI section (Section 4) with a comprehensive introduction to the PPI framework, referencing the original papers, and explaining how our estimator relates to the standard PPI formulation ($m_\theta + \Delta_\theta$). We also added detailed discussions of the assumptions and their roles.
>
> * In the experimental section, we explicitly link each synthetic regime to the corresponding theorem and discuss how the results corroborate the theory.
>
> * Added a new subsection (Section 5.2) comparing CP-POL with several state-of-the-art conformal methods (Split CP/APS, RAPS, TPS/Top-k, Jackknife+) on both synthetic and real-world data.
>
> **Where does Eq. (4.1) come from / is it in the PPI paper?** : Eq. (4.1) in our draft (the definition of $\widehat\Psi_i$ and the estimating equation $N^{-1}\sum \widehat\Psi_i(\widehat\theta)=0$) is algebraically equivalent to the $m_\theta+\Delta_\theta$ decomposition used in Angelopoulos et~al. (2023).  The Angelopoulos paper expresses the PPI test statistic as a plug-in measure computed on the unlabeled pool plus a labeled rectifier. Solving the root of our moment is the point-estimator version of that same object.  We made this equivalence explicit in the revised text and added the exact algebraic steps.
>
> **Sequential result clarified**: For the streaming theorem we added a paragraph giving the intuition for $V_T$, the martingale variance control, and for why we use the $\delta_T$ schedule to obtain time-uniform control.
>
> We hope these revisions address the reviewer’s concerns.

---

### Decision · Action_Editor_gXV9 · 2026-04-01

**Recommendation:** Accept as is

**Audience:**

Yes

**Audience Explanation:**

The results can be of interest for the community of ML researchers working on conformal prediction methods. In particular, the paper analyzes coverage of conformal prediction in practical settings and aligns with TMLR's interest in reliable and theoretically grounded ML.

**Claims And Evidence:**

Yes

**Claims Explanation:**

The paper provides theoretical results validated through simulations. In particular, the claims are supported by mathematical poofs, and the empirical results mirror the theoretical contributions.